# Inhibitory circuits control leg movements during *Drosophila* grooming

**Durafshan Sakeena Syed\*, Primoz Ravbar, Julie H Simpson\***

Neuroscience Research Institute and Department of Molecular, Cellular and Developmental Biology, University of California, Santa Barbara, Santa Barbara, United States

## eLife Assessment

Combining connectomics, optogenetics, behavioral analysis and modeling, this study delivers **important** findings on the role of inhibitory neurons in the generation of leg grooming movements in *Drosophila*. The results include **convincing** evidence that the identified neuronal populations are key in the generation of rhythmic leg movements, structured in distinct polysynaptic pathways articulating inhibition and disinhibition of antagonistic sets of motor neurons, as mapped from an electron microscopy volume of the ventral nerve cord, which orchestrate an alternation of flexion and extension. By analyzing limb kinematics upon experimentally silencing specific populations of premotor inhibitory neurons, together with computational modelling, the potential role of these neurons in rhythmic leg movement is shown. This work will be of interest to neuroscientists working in motor control and limbed locomotion.

**\*For correspondence:**
dsakeena@gmail.com (DSS);
jhsimpson@ucsb.edu (JHS)

**Competing interest:** The authors declare that no competing interests exist.

**Abstract** Limbs execute diverse actions coordinated by the nervous system through multiple motor programs. The basic architecture of motor neurons that activate muscles which articulate joints for antagonistic flexion and extension movements is conserved from flies to vertebrates. While excitatory premotor circuits are expected to establish sets of leg motor neurons that work together, our study uncovered an instructive role for inhibitory circuits — including their ability to generate rhythmic leg movements. Using electron microscopy data in the *Drosophila* nerve cord, we categorized ~120 GABAergic inhibitory neurons from the 13 A and 13B hemilineages into classes based on similarities in morphology and connectivity. By mapping their connections, we uncovered pathways for inhibiting specific groups of motor neurons, disinhibiting antagonistic counterparts, and inducing alternation between flexion and extension. We tested the function of specific inhibitory neurons through optogenetic activation and silencing, using high-resolution quantitative analysis of leg movements during grooming. We combined findings from anatomical and behavioral analyses to construct a computational model that can reproduce major aspects of the observed behavior, demonstrating that these premotor inhibitory circuits can generate rhythmic leg movements.

## Introduction

All animals with limbs face the challenge of coordinating their movements to achieve precise motor control. Despite a limited set of muscles in each limb, the nervous system produces multiple flexible actions to generate behavior. These movements rely on a balance of inhibition and excitation, although the specific circuitry remains unclear. In various insects, coordinated leg movements have been studied extensively during walking and grooming, revealing both the flexibility and stereotypy of action sequences (*Cruse, 1990*; *Berkowitz and Laurent, 1996a*; *Dürr and Ebeling, 2005*; *Schilling and Cruse, 2020*; *Büschges and Ache, 2025*). These studies highlight how insects provide a

powerful system to understand the circuit basis of limb coordination. *Drosophila* grooming involves coordinated, rhythmic leg movements to sweep the body and remove debris (*Ravbar et al., 2021*) with different actions prioritized sequentially. While sensory, command-like, and motor neurons (MNs) are known components of grooming circuits (*Seeds et al., 2014*; *Hampel et al., 2017*; *Zhang et al., 2020*; *Zhang and Simpson, 2022*; *Guo et al., 2022*; *Eichler et al., 2024*; *Yoshikawa et al., 2024*), the contribution of GABAergic inhibitory neurons has not been systematically investigated. We hypothesize that these neurons contribute to limb coordination and subroutine selection.

Leg movements during grooming require precise flexor-extensor coordination, controlled by MNs and premotor circuits. Flies use 14 intrinsic leg muscles and 3–5 body wall muscles (*Miller, 1950*; *Soler et al., 2004*; *Azevedo et al., 2024*), organized into antagonistic pairs. Around 70 excitatory MNs innervate these muscles (*Azevedo et al., 2024*; *Azevedo et al., 2020*; *Baek and Mann, 2009*; *Brierley et al., 2012*; *Phelps et al., 2021*) and recent connectomic mapping has revealed premotor neurons in the ventral cord neuromeres associated with each leg (*Lesser et al., 2024*; *Cheong et al., 2024*). There are around 622 local premotor interneurons (*Lesser et al., 2024*), suggesting complex control architectures. Studies in stick insects, locusts, cockroaches, and crustaceans have also revealed large numbers of interconnected pre-motor interneurons, some of which constitute central pattern generator circuits that shape rhythmic movements and interlimb coordination (*Büschges and Ache, 2025*; *Pearson, 1972*; *Bässler and Wegner, 1983*; *Büschges, 1998*; *Büschges and Manira, 1998*; *Mantziaris et al., 2020*; *Büschges, 2005*; *Ritzmann and Büschges, 2007*; *Büschges et al., 2011*; *Marder and Bucher, 2001*; *Marder et al., 2022*; *Calabrese and Marder, 2025*). In *Drosophila* larvae, circuits containing both excitatory and inhibitory interneurons generate rhythmic muscle contractions within segments and propagate peristaltic waves between segments for crawling (*Heckscher et al., 2012*; *Gjorgjieva et al., 2013*; *Kohsaka et al., 2014*; *Pulver et al., 2015*; *Fushiki et al., 2016*; *Zarin et al., 2019*). Together, these findings support the idea that premotor neurons may be a core component for generating and coordinating rhythmic motor control.

We investigate the role of inhibitory neurons in coordinating which leg muscles in adult *Drosophila* work together or antagonistically, and how they might produce rhythmic alternations. Neurons from a given lineage usually share a neurotransmitter, and there are 12 GABAergic hemilineages present in the ventral nerve cord (VNC; *Lacin et al., 2019*). We identified neurons from GABAergic 13 A and 13B hemilineages in a behavioral screen for grooming defects. Approximately 67±6 13 A neurons and 47±1 13 B neurons have been reported per hemisegment (*Lacin et al., 2019*; *Harris et al., 2015*). While activating all 13B and some 13 A neurons induces leg extensions (*Lacin et al., 2019*; *Soffers et al., 2025*), further investigation is required to understand the specific role of different subsets in leg coordination. The role of 13 A neurons has been unclear due to the lack of tools for exclusive genetic labeling.

Since grooming and walking (*Gowda et al., 2024*) require leg movements, we expect neural control circuits to overlap, particularly in flexion-extension alternation. MNs controlling similar muscles within a joint receive similar premotor inputs (*Lesser et al., 2024*). Descending neurons hypothesized to be involved in walking synapse onto premotor inhibitory neurons from several lineages (*Cheong et al., 2024*).

These central circuits also receive sensory feedback from leg proprioceptors (*Mendes et al., 2013*; *Tuthill and Azim, 2018*; *Chockley et al., 2022*; *Lee et al., 2025*). Recent connectome mapping and genetic tools provide opportunities to test how central and peripheral signals coordinate limb movement.

Muscle synergies describe groups of co-activated muscles, while motor primitives are elemental movement patterns that serve as building blocks of behavior (*Bernstein, 1967*; *Bizzi and Cheung, 2013*). Micro-stimulation of specific spinal cord regions in vertebrate models induced coordinated muscle contractions (*Sherrington, 1892*; *Sharrard, 1964*; *Ferrier and Yeo, 1881*; *Bizzi et al., 1991*; *Giszter et al., 1993*; *Lemay and Grill, 2004*; *Mussa-Ivaldi et al., 1994*), with co-stimulation leading to combinations of contractions, indicating how coordinated regulation by premotor circuits can simplify assembly of more complex movements (*Bizzi and Cheung, 2013*; *Mussa-Ivaldi et al., 1994*; *Kargo and Giszter, 2000*; *Lemay et al., 2001*). Additional evidence for synergies comes from electromyography, kinematics, neural recordings, and computational modeling (*Maier and Hepp-Reymond, 1995*; *Huesler et al., 2000*; *Ivanenko et al., 2003*; *Hart and Giszter, 2004*; *Yakovenko et al., 2011*; *Over-duin et al., 2012*; *Singh et al., 2018*; *Kutch and Valero-Cuevas, 2012*; *Capaday and van Vreeswijk,*

*2015*). In vertebrates, the neural circuits responsible for the coordinated limb movements include motor pools, a topographic map, and commissural neurons (*Romanes, 1951*; *Cruce, 1974*; *Tsuchida et al., 1994*; *Lanuza et al., 2004*; *Bellardita and Kiehn, 2015*; *Sengupta et al., 2021*; *Yang et al., 2023*). The activation of muscles controlling multiple joints (synergies) has been primarily described in terms of excitation (*Bizzi and Cheung, 2013*; *Sherrington, 1892*; *Sharrard, 1964*; *Ferrier and Yeo, 1881*; *Bizzi et al., 1991*; *Giszter et al., 1993*; *Lemay and Grill, 2004*). Muscle synergies may simplify motor control in insect walking and flight (*Berkowitz and Laurent, 1996a*; *Berkowitz and Laurent, 1996b*; *Zill et al., 2015*; *Zill et al., 2018*; *Sponberg et al., 2015*; *Ruthe et al., 2024*).

We hypothesize that similar synergies could simplify control of the rhythmic leg movements during grooming in flies and that inhibitory circuits play a critical role. We find that inhibitory neurons target different groups of MNs, providing an alternative way to construct muscle synergies. We demonstrate that normal activity of these inhibitory neurons is essential for the rhythmic coordination of leg flexion and extension during grooming. Knocking down inhibitory receptors on MNs in flies reduces locomotion speed (*Gowda et al., 2018*), but the specific inhibitory neurons involved remain unidentified. By analyzing limb kinematics in grooming flies and silencing specific 13 A and 13B neurons, we demonstrate their critical role in spatial and temporal limb coordination. Our findings suggest inhibitory circuits play a broader role in coordinated and rhythmic limb movements.

## Results

We describe how inhibitory 13 A and 13B neurons affect grooming. We categorize them by morphology and map their connectivity patterns. We present evidence for muscle synergies and a role for 13 A/B neurons in coordinating rhythmic limb movements. Finally, we generate a computational model to integrate and simulate our findings.

### Inhibitory interneurons in 13A and 13B hemilineages affect grooming behavior

Broad optogenetic activation of inhibitory neurons causes freezing, while activation of fewer neurons in the hemilineages 13 A and 13B results in reduced grooming behavior and poor leg coordination, including static over-extension of front legs in clean or dusted flies (*Figure 1D, E, E' and F*, *Figure 1— video 1*). We conclude that the critical neurons are located in the ventral cord because most Split GAL4 lines only express in the VNC (*Figure 1—figure supplement 1B and C*, *Figure 4—figure supplement 2A*). For the ones that do have some brain expression, optogenetic activation in decapitated flies still produces leg extension phenotypes (*Figure 1E-H*, *Figure 1—figure supplement 1A*).

Conversely, silencing these neurons locks the front legs in flexion (*Figure 1G, G' , and H*, *Figure 1— video 1*). Thus, activation or silencing of inhibitory neurons interferes with the alternation of flexion and extension required for dust removal and reduces grooming.

### Spatial mapping and connectivity patterns of premotor 13A neurons in leg motor control

The *Drosophila* nervous system develops from neuroblasts, each of which produces two hemilineages. These lineages arise from neuroblasts that divide briefly embryonically to produce primary neurons and later generate secondary neurons post-embryonically (*Lacin et al., 2019*; *Harris et al., 2015*; *Shepherd et al., 2016*; *Truman et al., 2004*; *Birkholz et al., 2015*; *Lacin and Truman, 2016*; *Marin et al., 2024*). We focus on the 13 A and 13B hemilineages. Using a serial section electron microscopy dataset (*Soler et al., 2004*), we identified 62 13A neurons and 64 13B neurons in the right front leg neuromere (T1-R) of VNC. Upon entering the neuropil, 13 A bundle divides into three sub-bundles, with five large neurons having extensive arbors, mostly in the ventral first sub-bundle of T1-R (*Figure 2—video 1*). Based on size, location, and connections, we hypothesize these represent early-born primary 13 A neurons. We also identified them in the left hemisegment distributed across three sub-bundles.

Lineages contain neurons with distinct shapes and functions. We used NBLAST (*Costa et al., 2016*), a computational tool designed to group neurons based on morphological similarities, to categorize 13 A and 13B neurons into distinct clusters (*Figure 2A*, *Figure 2—figure supplement 3*, *Figure 2— video 2*). Connectivity patterns were analyzed using automatic synapse detection algorithms

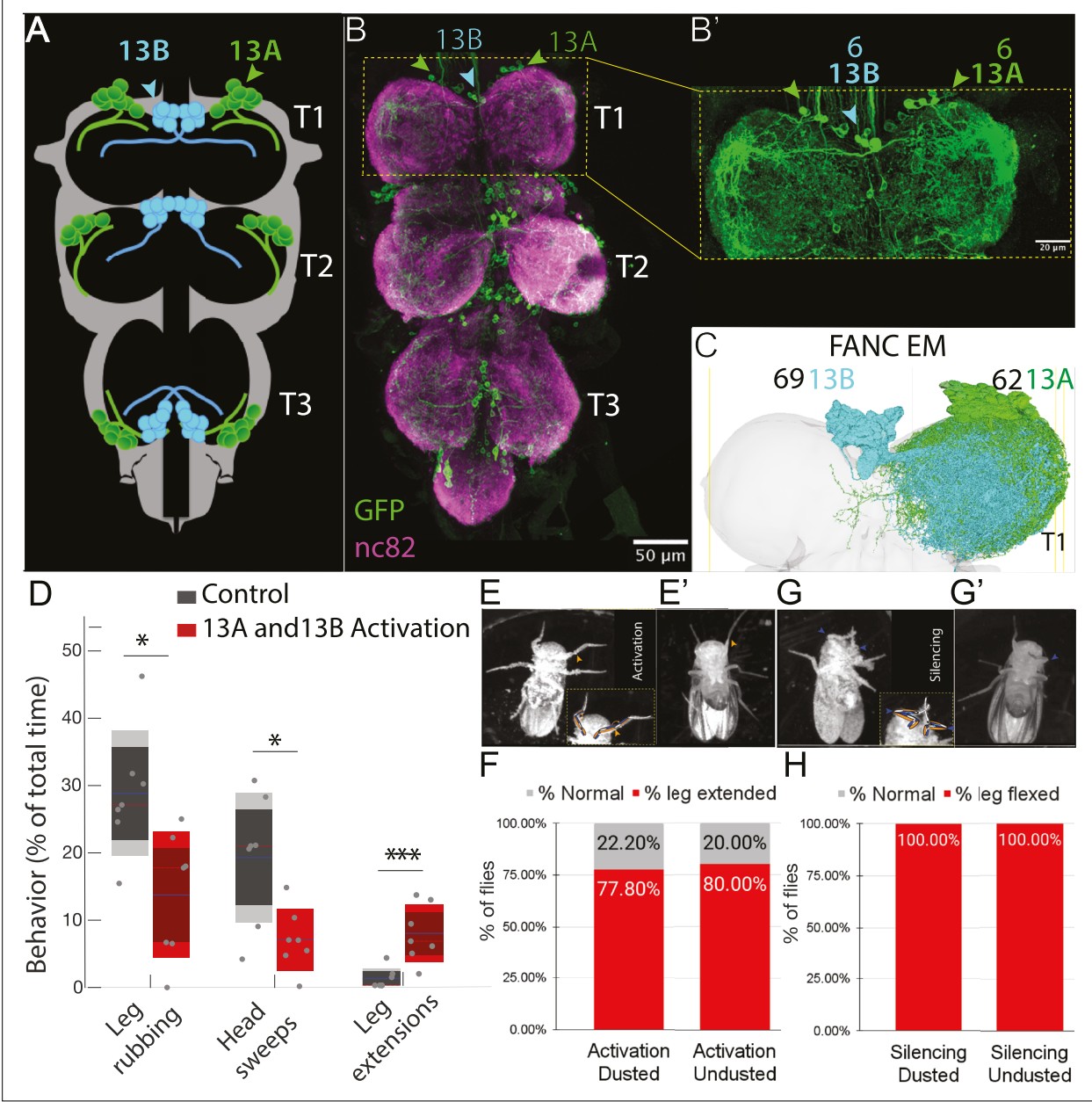

**Figure 1.** Anatomical Distribution and Behavioral Contributions of 13 A and 13B Hemilineages. Schematic showing segmental distribution of 13 A (green) and 13B (cyan) neurons across pro-, meta-, and meso-thoracic segments (**T1, T2, T3**) of VNC. Confocal image: Six GABAergic 13 A neurons (green arrowheads) and six 13B neurons (cyan arrowheads) in each VNC hemisegment, labeled with GFP (green) driven by *R35G04-GAL4-DBD, GAD-GAL4-AD*. Neuropil in magenta (nc82). Panel B' provides a zoomed-in view of T1 region. EM reconstructions: 62 13 A neurons (green) and 64 13B neurons (cyan) in right T1. Ventral side up. (**A**) Continuous activation of 13 A and 13B neurons labeled by *R35G04-GAL4-DBD, GAD-GAL4-AD* in dusted flies reduces front leg rubbing and head sweeps and induces unusual leg extensions. Control: *AD-DBD EMPTY SPLIT >UAS CsChrimson* (gray). Experiment: *R35G04-GAL4-DBD, GAD-GAL4-AD>UAS CsChrimson* (red). Box plots indicate the percentage of time dusted fly engaged in a given behavior over a 4-min assay (n=7). The solid blue line marks the mean, dark shading the 95% confidence interval, red dashed line the median, and light shading ± 1 standard deviation. ***p≤0.001, *p≤0.05. (**E-F**) Continuous activation of 13 A and 13B neuron subsets induces front leg extension in headless flies. (**E, E'**) Representative video frames showing headless flies (dusted and undusted) with extended front legs (orange arrowhead) following continuous optogenetic activation of neurons labeled with *R35G04-GAL4-DBD, GAD-GAL4-AD>UAS- CsChrimson*. Dashed box in E highlights the front legs; schematic illustrates the extended posture. (**F**) Quantification of leg extension phenotypes in dusted and undusted headless flies. Bar plots show the percentage of flies displaying leg extension (red) or a normal posture (gray). Percentages are calculated as the number of flies showing each posture divided by the total number of flies per condition. Dusted: n=9; undusted: n = 5. (**G–H**) Silencing 13 A and 13B neuron subsets locks front legs in flexion in headless flies. (**G, G'**) Representative video frames showing dusted and undusted headless flies with sustained front leg flexion following silencing of neurons labeled with *R35G04-GAL4-DBD, GAD-GAL4-AD>UAS* TNTe. Blue arrowheads indicate the flexed posture. (**H**) Quantification of leg flexion

*Figure 1 continued on next page*

*Figure 1 continued*

phenotypes in dusted and undusted headless flies. Bar plots show the percentage of flies displaying sustained flexion (red). All flies (100%) in both dusted (n=13) and undusted (n=9) conditions showed the phenotype. Also see *Figure 1—video 1*.

The online version of this article includes the following video and figure supplement(s) for figure 1:

**Figure supplement 1.** Expression pattern in the central nervous system of various lines used for behavior experiments.

**Figure 1—video 1.** Manipulating the activity of six 13 A neurons and six 13 A neurons in headless flies.

https://elifesciences.org/articles/106446/figures#fig1video1

---

(*Azevedo et al., 2024*). While 13 A neurons have many post-synaptic partners, their primary targets are MNs (*Figure 2—figure supplement 4*). Each morphological 13 A cluster connects to distinct sets of MNs. Clustering by MN connections correlated with NBLAST morphological clusters, as demonstrated by a cosine similarity matrix (*Figure 2*, *Figure 2—figure supplement 1*). While morphological and connectivity-based clusters align, exceptions include cluster 9 and the 13 A-3g, which have more diverse connections (*Figure 2B*, *Figure 2—figure supplement 1B6, H1-H9*). Our initial analysis used 13 A neurons in the right front leg neuromere (T1R). Comparison to a similar set on the left (*Lesser et al., 2024*) revealed similar numbers of neurons and cluster divisions (*Figure 2—figure supplement 2*).

Anatomical features of 13 A types suggest possible functional organization. Their dendrites occupy specific regions of VNC, suggesting common pre-synaptic inputs. Axons of 13 A neurons overlap with MN dendrites (*Figure 2A and B*, *Figure 2—video 3*), which are spatially segregated by the leg muscles they innervate, forming a myotopic map (*Baek and Mann, 2009*; *Brierley et al., 2012*). Cosine similarity between morphological clusters and MN targets indicates that the spatial position of 13 A neurons predicts which MN groups they connect to. Although 13 A neurons respect the myotopic organization of motor neurons, many of their connections span multiple MN groups. This shows that the 13 A spatial map could reflect premotor synergies (coordinated multi-joint control) rather than a strict one-to-one mapping between neurons and individual muscles.

Some 13 A neurons connect to multiple MNs across various leg segments; others target only a few. We classify these as 'generalists' and 'specialists'. We propose that the broadly projecting generalists are early-born primary neurons and that the specialists that target fewer MNs are later-born secondary neurons. This is consistent with the known developmental sequence of hemilineages, where early-born primary neurons typically acquire larger arbors and integrate across broader premotor and motor targets, whereas later-born secondary neurons often have more spatially restricted projections and specialized roles (*Baek and Mann, 2009*; *Brierley et al., 2012*; *Shepherd et al., 2016*; *Truman et al., 2004*; *Marin et al., 2024*). Our morphological clustering supports this idea: generalist 13As have extensive axonal arbors targeting motor neurons that control multiple leg segments, whereas specialist neurons are more narrowly tuned, connecting to a few MN targets within a segment. Thus, both their morphology and connectivity patterns align with the expectation from birth-order–dependent diversification within hemilineages. Four primary 13 A neurons (13A-10f-α, –9d-γ, –10g-β, and –10e-δ) are generalists (*Figure 2—figure supplement 1I5-I8 and I9*). Secondary neurons in clusters 1, 2, 4, 5, and 7 neurons are specialists, while clusters 6, 8, and 10 are generalists. Clusters 3 and 9 contain a mix. Specialist neurons tend to target MNs in a more restricted, segment-specific manner, reflecting a myotopic map, whereas generalist inhibitory neurons target specific MN groups across multiple leg segments, reflecting premotor synergies—coordinated group of muscles that work together. These overlapping patterns suggest that 13As form a spatial map organized by muscle synergies, rather than a strict one-to-one myotopic map. Together, these findings indicate that 13 A neurons constitute a spatially organized premotor map, where morphology and connectivity jointly predict the recruitment of both broad, multi-joint synergies and restricted, joint-specific motor outputs.

## Connectivity motifs for coordinated control

The adult *Drosophila* leg consists of five joints, named according to the segments they connect: body-wall/thoraco-coxal (Th-C), coxa–trochanter (C-Tr), trochanter–femur (Tr-F; fused in the adult), femur–tibia (F-Ti), tibia–tarsus (Ti-Ta). Each joint is powered by opposing flexor and extensor muscles that transmit force through tendons (*Soler et al., 2004*). The proximal joints, Th-C and C-Tr, mediate leg protraction–retraction and elevation–depression, respectively (*Büschges and Ache, 2025*). The medial

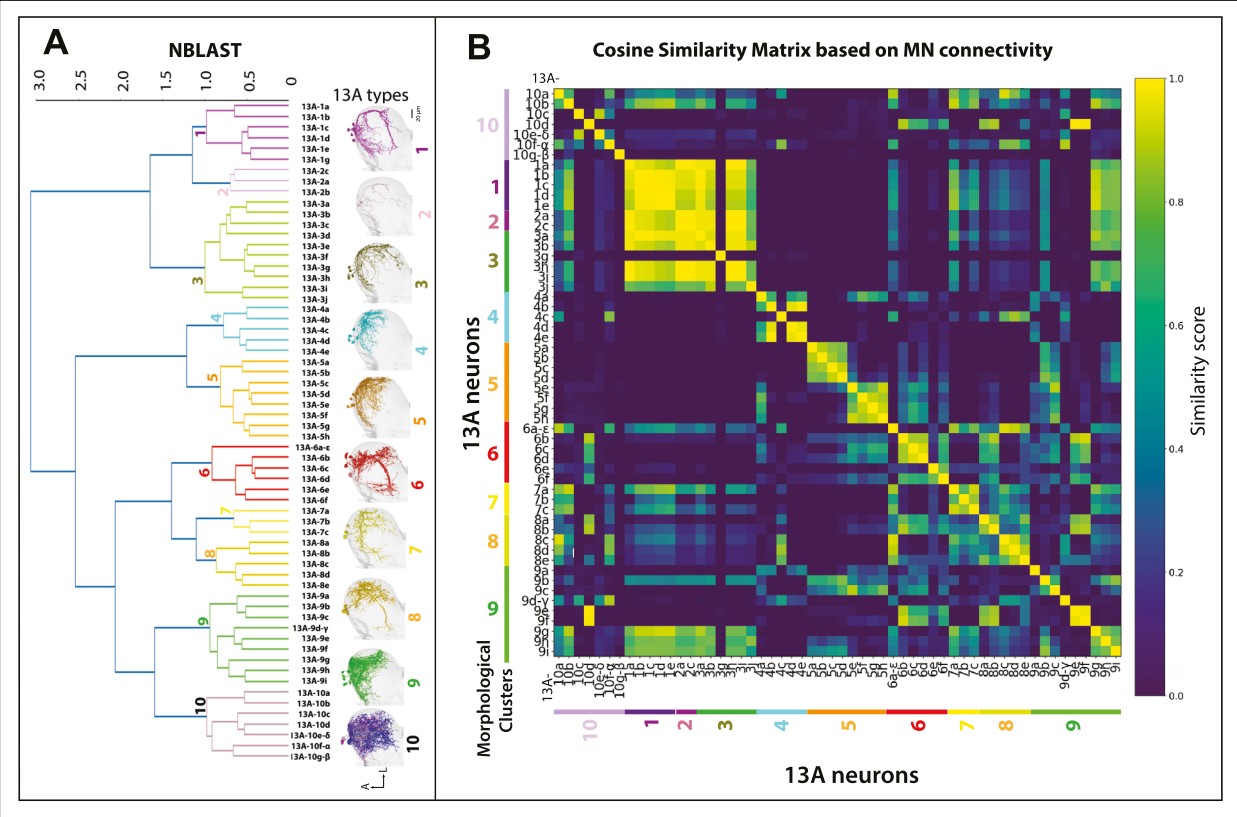

**Figure 2.** Spatial map of premotor 13 A neurons correlates with their connections to motor neurons (MNs). *Hierarchical clustering of 13 A hemilineage.* Clustering of 13 A neuron types in the right T1 segment was performed using NBLAST, resulting in identification of 10 morphological groups or clusters. EM reconstructions of distinct 13 A clusters are shown. Neurons are named based on morphological clustering. For example, all neurons in the 13 A-3 cluster have similar morphology, with 10 neurons labeled as 13 A-3 (**a–j**) (olive). Images of each 13 A neuron are shown in *Figure 2—figure supplement 1*. Also see *Figure 2—video 2*. A=anterior, L=Lateral. Ventral side is up. (**A**) *Cosine similarity graph showing pairwise similarity between 13 A neurons based on their MN connectivity patterns.* 13 A neurons are organized based on anatomical clusters obtained with NBLAST as described above. It depicts a correlation between the anatomy of 13 A neurons and their connections to MNs. For example, 13 A-1a, –1b, –1 c, –1d (cluster 1) connect to the same set of MNs, therefore have high cosine similarity with each other (as seen across the diagonal). The graph also gives insights into groups of 13As that control similar muscles. For example, cluster 1 neurons have high cosine similarity with cluster 3 13 A neurons (while, 3 g neuron is an exception).

The online version of this article includes the following video and figure supplement(s) for figure 2:

**Figure supplement 1.** Spatial map and connectivity of premotor 13 A neurons.

**Figure supplement 2.** Connectivity matrix of 13 A neurons and the motor neurons showing left-right comparison and spatial map.

**Figure supplement 3.** Anatomical classification of 13B neurons.

**Figure supplement 4.** Neurons downstream of a primary 13A-10f-α neuron.

**Figure 2—video 1.** 13 A neurons in the right front leg neuromere (T1R).

https://elifesciences.org/articles/106446/figures#fig2video1

**Figure 2—video 2.** 13 A morphological clusters.

https://elifesciences.org/articles/106446/figures#fig2video2

**Figure 2—video 3.** 13 A cluster 6 neurons (red) and downstream Tibia extensor MNs (feti and seti, green).

https://elifesciences.org/articles/106446/figures#fig2video3

joint, F-Ti, acts as the principal flexion–extension hinge and is controlled by large tibia extensor motor neurons and flexor motor neurons (*Soler et al., 2004*; *Azevedo et al., 2024*; *Baek and Mann, 2009*; *Brierley et al., 2012*; *Lesser et al., 2024*). By contrast, distal joints such as Ti-Ta and the tarsomeres contribute to fine adjustments, grasping, and substrate attachment (*Azevedo et al., 2024*).

We analyzed the connectivity of 13 A/B neurons synapsing onto MNs of the medial (F-Ti) joint (*Figure 3*). Interconnections between 13 A neuron types suggest a role in generating flexor-extensor antagonism. 13 A neurons synapsing onto extensor MNs also inhibit 13 A neurons targeting flexors,

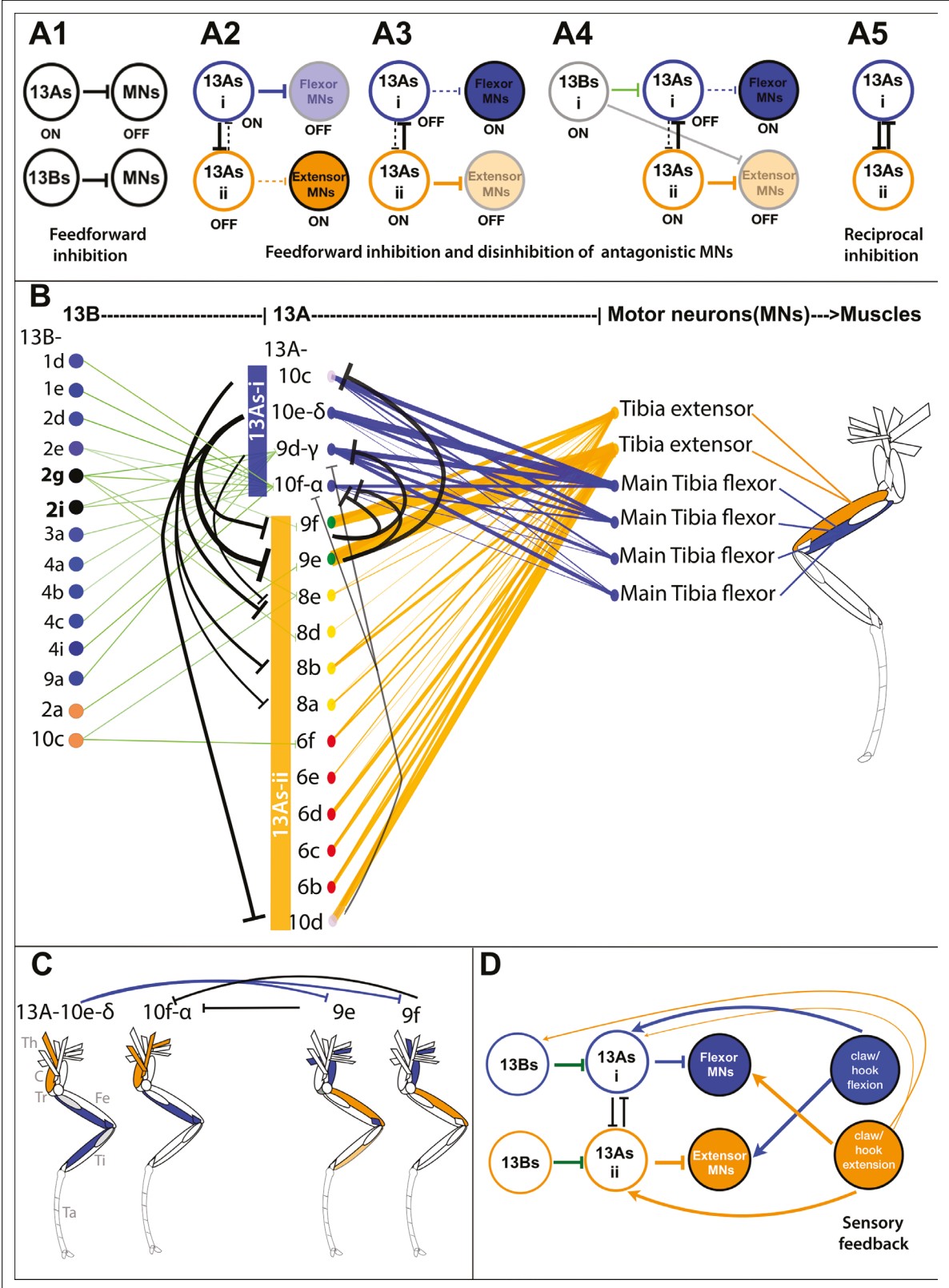

**Figure 3.** Inhibitory circuitry for antagonistic muscle control. *Schematic of inhibitory circuit motifs.* (**A1**) Feedforward inhibition by 13 A/B neurons. (**A2**) Flexor inhibition and extensor disinhibition: 13As-i inhibit flexor MNs and disinhibit extensor MNs by inhibiting 13As-ii. (**A3**) Extensor inhibition and flexor 1296 disinhibition: 13As-ii inhibit extensor MNs and disinhibit flexor MNs by inhibiting 13As-i. (**A4**) 13B mediated disinhibition: 13Bs disinhibit MNs by targeting premotor 13As, while some also directly inhibit antagonistic MNs. (**A5**) Reciprocal inhibition among 13 A groups that inhibit

*Figure 3 continued on next page*

*Figure 3 continued*

antagonistic MNs may induce flexor-extensor alternation. *Connectivity matrix*: Inhibitory connections regulating antagonistic MNs of the medial joint. Leg schematic shows tibia extensor (orange) and flexor (blue) muscles, innervated by respective MNs. Flexor-inhibiting 13 A neurons (13As-i) in blue, and extensor-inhibiting 13As (13As-ii) in orange. The thickness of edges between nodes is determined by the number of synapses. Node colors were assigned based on the type of neurons, with specific colors denoting different subtypes of 13 A/B neurons and MNs. *Feedforward inhibition*: Primary neurons (13A-10f-α, 9d-γ, and 10e-δ) and 13 A-10c (13As-i) connect to tibia flexor MNs (blue edges), making a total of 85, 219, 155, and 157 synapses, respectively. Twelve secondary 13As ii inhibit tibia extensor MNs (orange edges), with strong connections from 13 A-9f, –9e, and –10d totaling 188, 275, 155 synapses, respectively. *Reciprocal inhibition*: Three neurons from 13As-i inhibit six from 13As ii, with 13A-10e-δ connecting to 13A-9f (19 synapses), -9e (31), and -10d (14). 13A-10c connects to 13A-8a (6), -8b (12), and -8e (5). 13A-9d-γ connects to 13A-8e (8). Conversely, three from 13As-ii inhibit two neurons from 13As i, with 13A-9f connecting to 13A-10f-α (25) and -9d-γ (6), and 13A-10d connecting to 13A-10f-α (8), -9d-γ (7), and -10e-δ (15). 13A-9e connects to 13A-10f-α (21) and -10c (47) (black edges). *Disinhibition by 13B neurons*: 13B connects to 13As-i (13A-10f-α and -9d-γ) (totaling 78 and 50 synapses) (green edges), disinhibiting flexor MNs. 13B-2g and –2i also directly inhibit tibia extensor MNs. Reciprocal inhibition for multi-joint coordination: Primary 13As (10e-δ and 10f-α) target a combination of proximal (sternotrochanter, tergotrochanter, trochanter extensor, tergopleural promotor), medial (tibia flexor), and distal (tarsus depressor) MNs, while secondary 13As (9e and 9f) target antagonist MNs including sternal posterior rotator, pleural remotor abductor, and tibia extensor. Reciprocal connections between them indicate that generalist 13As coordinate multi-joint muscle synergies through inhibition of antagonistic motor groups. Leg schematic shows the muscles innervated by the corresponding MNs in various leg segments (Th = thorax, C = coxa, Tr = trochanter, Fe = femur, Ti = tibia, Ta = tarsus). (Data for 13A-MN connections are shown in *Figure 2—figure supplement 1I9, I6, I7, H9, H4, and H5; 13A-13A* connections shown in *Figure 3—figure supplement 1C*). *Proprioceptive feedback*: Sensory feedback from proprioceptors onto reciprocally connected 13As could turn off corresponding MNs and activate antagonistic MNs. Flexion-sensing proprioceptors target extensor MNs and 13As-i that inhibit tibia flexor MNs. Extension-sensing proprioceptors target tibia flexor MNs and two 13As (13As-ii) that inhibit extensor MNs. Claw extension neurons also connect to 13A-δ. One 13B that disinhibits flexor MNs also receives connection from extension-sensing proprioceptors. Also see *Figure 3—figure supplement 3*.

The online version of this article includes the following figure supplement(s) for figure 3:

**Figure supplement 1.** Disinhibition matrix.

**Figure supplement 2.** Neurons Downstream of a 13B Neuron (13B-4i).

**Figure supplement 3.** Sensory feedback onto inhibitory 13 A neurons and motor neurons.

and vice versa (*Figure 3A2, A3 and B*). These redundant circuits could ensure that at a given time point, either extensor or flexor is active.

These 13 A groups are also reciprocally connected to each other (*Figure 3A4, A5 and B*), providing a mechanism for alternation between flexion and extension over time. This organizational motif applies to multiple joints within a leg as reciprocal connections between generalist 13 A neurons suggest a role in coordinating multi-joint movements in synergy (*Figure 3C*).

We did not find any correlation between the morphology of premotor 13B and motor connections, but there is a topographically restricted output based on their 13 A premotor targets (*Figure 2—figure supplement 1*, *Figure 3—figure supplement 1B*). 13B neurons connect to 13 A neurons targeting either flexor or extensor MNs (*Figure 3—figure supplement 1B* and *Figure 3—figure supplement 2*). Two specific 13B neurons inhibit both extensor MNs and disinhibit flexor MNs, playing a dual role. Twenty-four 13B neurons from clusters 1–4 target 13 A neurons.

Interconnections between 13 A and 13B neurons reveal additional inhibitory motifs that could mediate movement of other joints or multiple joints synergistically. 13B neurons disinhibit MNs by inhibiting premotor 13Bs or 13As. For example, 13B-4h inhibits 13B-2i, a generalist premotor neuron targeting proximal flexor and medial extensor MNs (*Figure 3—figure supplement 1A, E, F*), while preventing disinhibition of antagonistic MNs (*Figure 3B*, *Figure 3—figure supplement 1B*). Similarly, 13As could disinhibit antagonistic MNs by inhibiting premotor 13Bs. For example, primary 13A-10g-β connects to 16 13B neurons, disinhibiting proximal extensor MNs while inhibiting proximal/medial flexors (*Figure 3—figure supplement 1A, D, F*).

Together, inhibitory interconnections among 13 A neurons, and between 13 A and 13B neurons, may coordinate alternating activity in antagonistic muscle groups.

## Proprioceptive feedback to 13A neurons

We identified connections from position-sensing proprioceptors to primary 13 A neurons. These could provide sensory feedback. Claw neurons detect position, while hook neurons sense movement direction (*Mamiya et al., 2018*, *Agrawal et al., 2020*, *Chen et al., 2021*). We examined reconstruction of proprioceptive neurons (*Phelps et al., 2021*) and found multiple connections from flexion-sensing claw and hook neurons onto the main neurite of 13A-10f-α, which targets tibia flexor MNs (*Figure 3D*,

*Figure 3—figure supplement 3*). Similar connections were observed onto 13A-10e-δ, –9d-γ. Recent connectome analysis showed that flexion sensing proprioceptors send direct excitatory feedback to tibia extensor MNs and indirect inhibitory feedback to flexor MNs (*Lee et al., 2025*). Thus, flexion-sensing proprioceptors could activate primary 13 A neurons to inhibit tibia flexor MNs, while directly activating extensor MNs. Similarly, claw extension neurons connect to two 13 A neurons that target tibia extensor MNs, while directly connecting to flexor MNs. Since these two groups of 13 A neurons receive proprioceptive feedback and reciprocally inhibit each other, they could drive flexion-extension alternation.

## Behavioral evidence for muscle synergies during grooming

Dusted flies use their legs to perform precise grooming actions, involving repeating patterns of body sweeps and leg rubs (*Ravbar et al., 2021*; *Seeds et al., 2014*). Quantifying these movements using machine vision methods (DeepLabCut) (*Mathis et al., 2018*) reveals synchronized changes in angular velocity across multiple leg joints (*Figure 4A*). During leg rubbing, proximal and medial joints within a given leg move predominantly in sync, as indicated by a minimal lag in their angular velocities (*Figure 4A'*), although they occasionally move asynchronously during head sweeps (*Figure 4A''*). This coordination shows the presence of muscle synergies, and generalist premotor interneuron connectivity could be how these synergies are implemented.

## 13A neurons affect limb coordination during grooming

Half of the 13 A population expresses a transcriptional factor, Dbx40. We used Split GAL4 combinations to target smaller subsets, intersecting with a GAD line and Dbx to manipulate a small subset of 13 A neurons.

These 13 A neurons (*Figure 4B*, *Figure 1—figure supplement 1B*) target MNs controlling multiple joints, including proximal (Th-C) (Sternotrochanter extensor, tergotrochanter, tergoplural promotor body wall muscles, and trochanter extensor MNs), medial (F-Ti) joint extensor/flexor MNs, and distal (Ti-Ta) joint tarsus extensor MN (*Figure 4—figure supplement 1A1-1A3*). Based on their connectivity, we hypothesized that silencing 13 A neurons, and thereby removing inhibition from these MNs, would lead to exaggerated or mistimed movements at proximal and medial joints and improper positioning of the distal tarsus. Specifically, proximal joints may show excessive retraction and elevation, medial joints may display uncoordinated flexion/extension, and distal tarsus movements may become poorly targeted or overshoot, resulting in reduced inter-leg spacing and inefficient, uncoordinated grooming strokes. Conversely, activating 13As would suppress MN activity, reducing leg rotation and joint movements, limiting tibia bending and tarsus contact, and thereby impairing coordinated grooming.

To test the function, we manipulated 13 A neuron activity in intact, behaving animals, where motor output and sensory feedback interact continuously. This allowed us to probe their role within the intact sensorimotor system. Given their connectivity with both motor neurons and proprioceptive inputs, 13 A neurons likely contribute to movement generation and its modulation by feedback. Therefore, the behavioral outcomes observed in our assays reflect their integrated role in motor control.

We measured changes in extension and flexion with joint positions, angles, and inter-leg distances. Activating or silencing six 13 A neurons reduced total grooming time and disrupted joint positions in dusted flies (*Figure 4—figure supplement 1E'–1F'*). Silencing 13 A neurons reduced mean distance between F-Ti joints of the front legs (*Figure 4C*) and decreased the frequency of extension-flexion cycles (*Figure 4F*), indicating that 13 A activity is required to sustain rhythmic motor output. Continuous activation of these neurons also showed a trend toward reduced frequency (although not statistically significant after correction) and increased variance suggesting that persistent activity makes the leg movements slower and more variable (*Figure 4H*).

To test whether these effects were specific to grooming behavior, we also examined walking bouts in these dusted flies. Silencing 13 A neurons did not affect the frequency of extension–flexion cycles during walking in dusted flies (*Figure 4—figure supplement 1G*). However, two spatial features were altered: the distance between the tarsal tips of the front legs and the distance between the tibia–tarsus joints were both reduced, indicating subtle effects on front limb posture during walking (*Figure 4—figure supplement 1G*).

Silencing specific 13 A neurons in dusted flies disrupted both spatial and temporal features of grooming, highlighting their necessity in producing precise grooming actions.

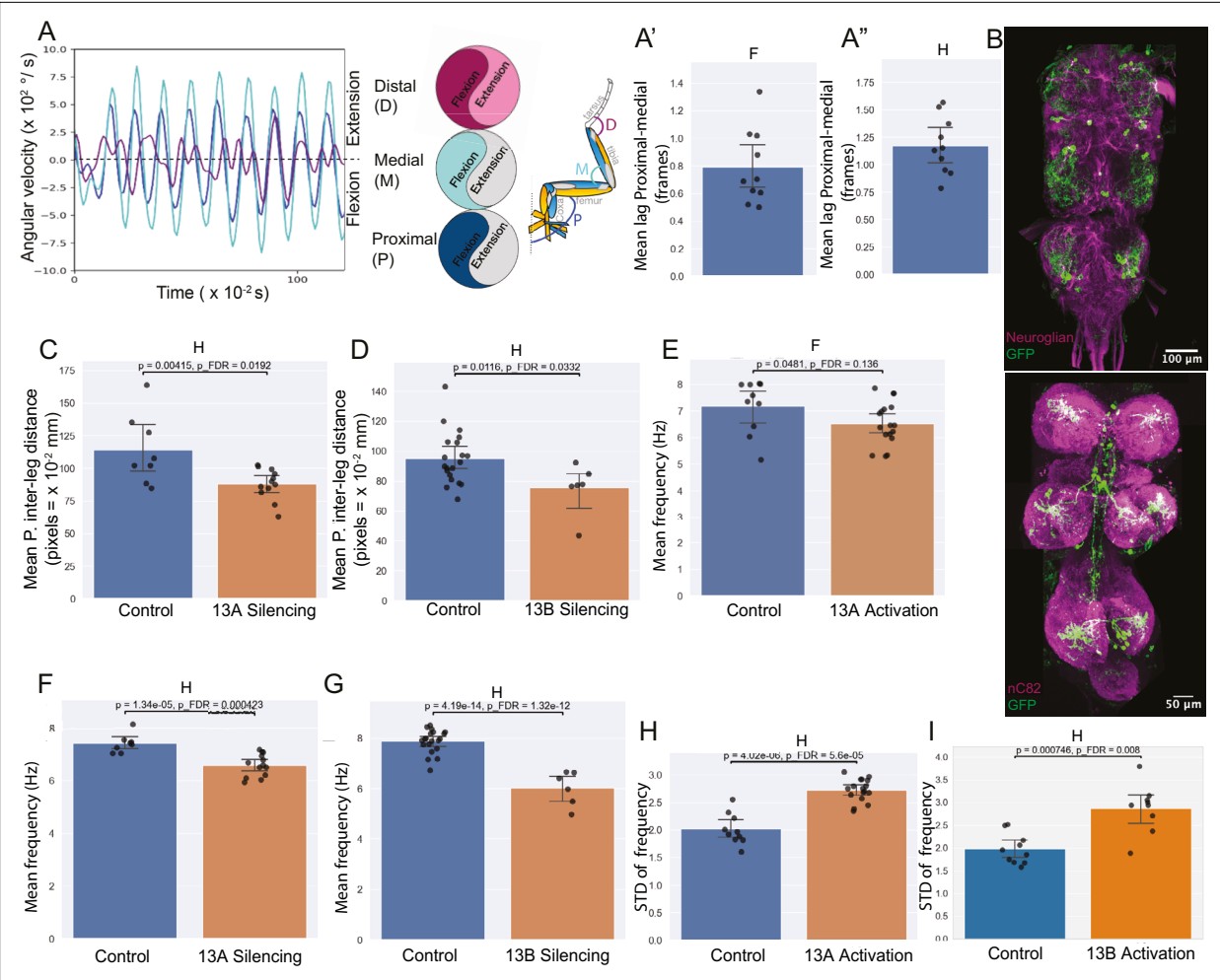

**Figure 4.** 13 A and 13B neurons are required for leg coordination during grooming. (**A-A"**) *Intra-joint coordination and muscle synergies*. Angular velocities of proximal (P, blue) and medial (M, cyan) joints predominantly move synchronously, while distal (D, purple) can move in or out of phase during leg rubbing. The schematic (right) indicates the corresponding joint angles. (**A'-A"**) The proximal and medial joint movements within a leg occur effectively in phase, with a mean lag of ~0.8 frames (8ms) during leg rubbing (A') and during head grooming sweeps (A"). Bar plots show the lag; each dot indicates one animal. Frame = 10ms. Neuronal labeling of 13A and 13B neurons. Top: Confocal image of six Dbx positive 13 A neurons per hemisegment labeled by GFP using *R35G04-GAL4-DBD, Dbx-GAL4-AD* in VNC. Neuroglian (magenta) labels axon bundles. Bottom: Confocal image of three 13B neurons per hemisegment labeled by GFP using *R11B07-GAL4-DBD, GAD-GAL4-AD*. Nc82 (magenta) labels neuropil. (**C–I**) Effects of neuronal activity manipulation in dusted flies. Silencing and activation of 13 A neurons in dusted flies using *R35G04-GAL4-DBD, Dbx-GAL4-AD with UAS Kir or UAS CsChrimson*, respectively (n=12 silencing, n=19 activation). Control: *AD-GAL4-DBD EMPTY SPLIT* with *UAS Kir or UAS CsChrimson*. For 13B neurons, *R11B07-GAL4-DBD, GAD-GAL4-AD* with *UAS GtACR1*, or *UAS CsChrimson*, respectively (n=7 silencing, n=9 activation); control: *AD-GAL4-DBD EMPTY SPLIT* with *UAS GtACR1 or UAS CsChrimson*. Each panel compares control (blue) and experimental (orange) groups. Each dot represents the mean feature value for a single fly. Bars indicate the group mean, and whiskers represent the 95% confidence interval of the group mean. *P*-values (raw and false discovery rate [FDR]–corrected) are shown above each panel. (**C–D**) *Proximal inter-leg distance*: Silencing of 13 A (**C**) or 13B (**D**) neurons during head grooming reduces the distance between the femur-tibia joints of the left and right front legs. (**E–I**) *Frequency modulation*: Silencing 13 A or 13B neurons reduces mean frequency of proximal joint oscillations in dusted flies. (**F, G**). Activation of 13 A neurons reduced frequency, although this change did not survive FDR correction. However, continuous activation of 13 A and 13B neurons increased variability in frequency. (**H, I**). Mean of the per-animal standard deviation (STD) that reflects variability or spread of data is shown.

The online version of this article includes the following figure supplement(s) for figure 4:

**Figure supplement 1.** 13 A neurons regulate leg coordination during grooming.

**Figure supplement 2.** Two Dbx-positive 13 A neurons are involved in leg coordination during grooming in dust-covered flies.

**Figure supplement 3.** Neuronal labeling.

Furthermore, optogenetic manipulation of two different Dbx-positive 13 A neurons also disrupted grooming, reducing joint position precision and mean frequency, without significantly affecting maximum angular velocity during grooming (*Figure 4—figure supplement 2*). Thus, 13 A neurons regulate leg coordination during grooming.

## 13B neurons affect limb coordination during grooming

Our experiments demonstrated that silencing or activating 13B neurons reduced grooming. Connectome data revealed that 13B neurons disinhibit groups of motor neurons. Most 13B neurons appear to act indirectly in the connectome—by contacting inhibitory neurons, potentially producing a net excitatory effect—while a subset makes direct contacts onto motor neurons. We generated a split GAL4 line that labels three 13B neurons (*Figure 4B*, *Figure 4—figure supplement 3*, and *Figure 1—figure supplement 1D*): two inhibit a primary 13 A neuron (13A-9d-γ) which targets proximal extensor and medial flexor MNs (*Figure 4—figure supplement 3C*), and one is premotor, directly inhibiting both proximal and tibia extensor MNs. Together, these 13B neurons could disinhibit proximal extensor and medial flexor MNs while inhibiting medial extensor MNs.

Activating or silencing these three 13B neurons in dusted flies also reduced grooming and resulted in joint positioning defects (*Figure 4D*, *Figure 4—figure supplement 3D', E'*). Silencing 13B neurons decreased proximal inter-leg distance (*Figure 4D*). Continuous activation of 13B neurons often resulted in one leg being locked in flexion while the other leg remained extended, perhaps indicating contribution from unknown left-right coordination circuits.

Manipulating 13B neuron activity also affects temporal aspects of grooming. Optogenetic silencing of 13B neurons in dusted flies strongly decreased mean frequencies of extension-flexion cycles of all joints (*Figure 4G*), while activation resulted in increased variability (*Figure 4I*). Both silencing and continuous activation decreased total time spent in anterior grooming; however, the bout duration of head sweeps increased upon silencing, and that of leg rubs slightly reduced in dusted flies (*Figure 4—figure supplement 3F-G'*).

Together, 13 A and 13B neurons contribute to both spatial and temporal coordination during grooming.

## Activation of inhibitory neurons induces rhythmic leg movements

Connectome analysis revealed that inhibitory 13 A and 13B neurons frequently synapse onto 13 A premotor neurons. Thus, activation of these 13 A or presynaptic 13B neurons should inhibit postsynaptic 13 A neurons, releasing activity in MNs and promoting movement. Consistent with this connectivity, we observe that optogenetic activation of specific 13 A and 13B neurons triggers grooming movements (*Figure 5D*, data not shown).

We also examined whether the timing of 13 A activation could influence frequency of these rhythmic grooming actions. Anterior grooming actions in dusted flies are rhythmic, where leg rubbing and body sweeps typically occur at a median frequency of ~7–8 Hz. One complete extension and flexion cycle, representing one sweep or leg rub, lasts ~140ms, with 70ms extension and 70ms flexion phases (*Figure 5B and B'*). Connectivity analysis suggests that specific 13 A neurons would be tuned to induce extension and others induce flexion, with reciprocal inhibition potentially generating rhythmicity. We optogenetically activated specific 13 A neurons using 70ms on and off light pulses to mimic the flexion-extension cycle in clean flies. This indeed induced grooming (anterior and posterior) and walking (*Figure 5C-D*, *Figure 5—video 1*). The frequency and maximum angular velocity of proximal joint movements during these induced behaviors closely matched those observed in dust-induced grooming (*Figure 5E and F*).

To test whether the induced rhythm was locked to the stimulation period, we delivered optogenetic stimulation with equal on/off pulses of 10ms (50 Hz), 50ms (10 Hz), 70ms (~7 Hz), 110ms (~4.5 Hz), and 120ms (~4 Hz) and compared the mean frequency of proximal joint cycles across conditions. Frequencies did not significantly differ across stimulation paradigms (*Figure 5—figure supplement 1*), suggesting that pulsed activation triggers the circuit's intrinsic rhythm rather than precisely pacing it.

We note that CsChrimson has relatively slow off-kinetics, which may limit the temporal precision of optogenetic control. Nevertheless, across a wide range of stimulation frequencies, optogenetic activation elicited alternating leg movements that were consistent with normal grooming behavior.

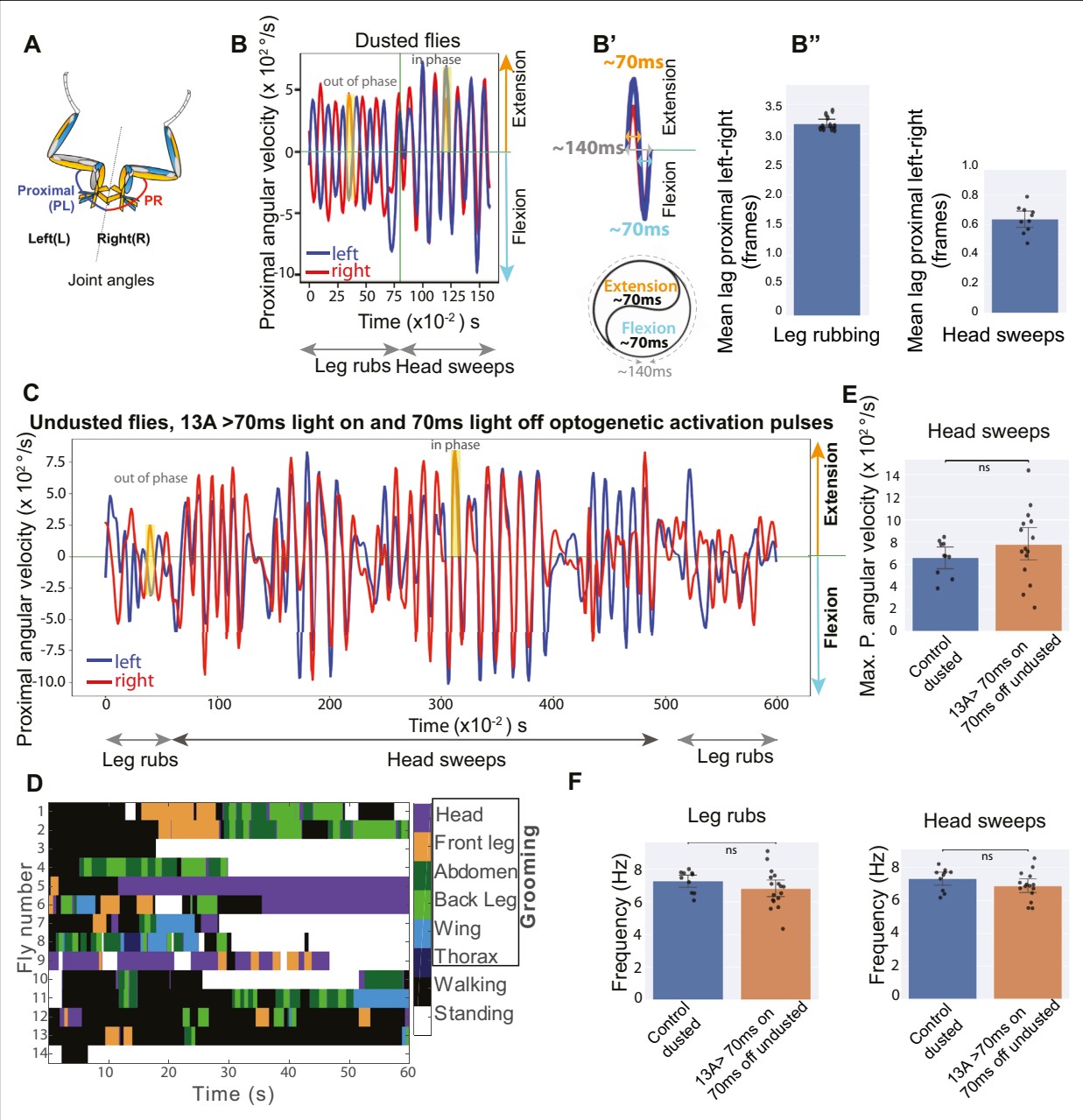

**Figure 5.** Pulsed activation of 13 A neurons triggers rhythmic actions in clean undusted flies. Schematic showing proximal joint angles of left (PL) and right (PR) legs. *Left-right coordination and muscle synergies during anterior grooming.* Dusted flies perform alternating leg rubs and head sweeps. Proximal joint angular velocities are shown. PL (blue) and PR (red) joints move anti-phase during leg rubs and in-phase during head sweeps (highlighted yellow box). Positive values indicate extension, and negative indicates flexion. (**B'**) Each flexion and extension cycle lasts ~140 ms, with each phase around 70 ms. (**B''**) Mean lags between proximal joints of the left and the right legs during leg rubbing and head sweeps. High lag during leg rubbing (left panel) indicates out-of-phase movement, and low lag during head sweeps (right) indicates in-phase movement. Bar plots show the lag; each dot indicates one animal. Frame = 10 ms. (**C–F**) Effect of optogenetic activation using 70 ms on and 70 ms off pulses in specific 13 A neurons (*R35G04-DBD, Dbx-GAL4-AD>UAS CsChrimson*) in undusted flies. Angular velocity of PL and PR leg joints shows anti-phase leg rubs and sustained in-phase head sweeps, with light pulses active from time = 0. Behavioral ethogram showing various grooming actions (head, front leg, abdomen, back leg, wing, thorax) and walking triggered by 70 ms on and 70 ms off pulsed activation of 13 A neurons in undusted flies, with light pulses on from time = 0. Maximum angular velocity of proximal joints during head sweeps upon pulsed 13 A activation in undusted flies is comparable to that observed in dusted flies. (**A**) The frequency of proximal joint movements during leg rubbing (left) and head sweeps (right) induced by 13 A pulsed activation is also similar between dusted and undusted flies. *Control flies: AD-DBD EMPTY SPLIT >UAS-CsChrimson, dusted; Experimental flies: R35G04-DBD, Dbx-GAL4-AD>UAS-CsChrimson, undusted. Light pulses were delivered at 70 ms on / 70 ms off. Each panel compares control (blue) and experimental*

*Figure 5 continued on next page*

*Figure 5 continued*

(orange) groups. Each dot represents the mean feature value for a single fly. Bars indicate the group mean, and whiskers represent the 95% confidence interval of the group mean. p-Values (raw and false discovery rate [FDR]–corrected) are shown above each panel.

The online version of this article includes the following video and figure supplement(s) for figure 5:

**Figure supplement 1.** Effect of varying optogenetic stimulation period on proximal joint cycle frequency.

**Figure 5—video 1.** Activation of six 13 A neurons with 70 ms on and 70 ms off pulses in undusted flies induces grooming and walking behaviors. https://elifesciences.org/articles/106446/figures#fig5video1

This experimental evidence shows that 13 A neurons can generate rhythmic movements, reinforcing their role in coordinating grooming behavior.

## A computational model of inhibitory circuits in coordinating grooming actions

The inhibitory circuits connecting to MNs are complex and genetic reagents to target their individual components are limited, so figuring out how each component contributes to leg coordination experimentally is challenging. We developed a neural computational model based on anatomical connectivity to explore potential circuit functions. Several studies have employed (more complex) neural models to show how oscillatory behaviors, such as walking, can emerge without an explicit need for a central controller, for example CPG (*Schilling and Cruse, 2020*; *Schilling and Cruse, 2023*). We modeled groups of functionally related neurons. For example, 13As that are inhibiting each other represent two groups modeled as two nodes – *Figure 6A* shows such a circuit for a single joint. This approach is loosely inspired by *Jovanic et al., 2016*. Since we are not modeling individual neurons, the network does not involve spiking neurons but rather 'rate based' units. The 'synaptic weights' of the model network correspond to the number of synapses obtained empirically from the connectome (*Figure 6A*).

The neural network controls movements of virtual front legs of an agent, where each leg is simplified to a 2D configuration of three segments. A pair of antagonistic 'muscles' controls each of the three 'joint' angles on each leg. These pairs of muscles determine the angular velocities of each 'joint'. Thus, each leg receives inputs from six virtual MNs. These MNs receive descending excitatory inputs and inhibitory inputs from two 13 A nodes (inhibiting the MNs of flexors and extensors). The states of the muscles (amounts of extension or flexion) are sensed by sensory neurons (SNs) that provide feedback to the 13As and to MNs. As the legs move, the most distal 'joint' removes the virtual dust (*Figure 6I*). Legs must also spend some time in proximity to each other to remove the 'dust' from themselves (this constraint forces the legs to coordinate with each other.).

The sensory input to the model circuit is a function of the distribution of 'dust' remaining on the 'body' - the environment—represented by the green areas in *Figure 6I* and *Figure 6—video 1*. We use a small recurrent neural network (RNN) that transforms the distribution of the 'dust' into excitatory inputs for the 13 A network. This simple RNN, consisting of 40 units, is a 'black box' used to provide the 13 A circuits with excitatory sensory inputs so that the agent can respond to the changing environment. Other inputs to the 'black box' include a copy of motor neuron's activation levels ('efference copy') and the amount of dust accumulated on the legs.

But the 13 A circuitry can still produce rhythmic behavior even without those excitatory inputs from the 'black box' (or when the inputs are set to a constant value), although the legs become less coordinated (because they are 'unaware' of each other's position at any time). Indeed, when we refine the model (with the evolutionary training) without the 'black box' (using a constant input of 0.1), the behavior is still rhythmic although somewhat less sustained (*Figure 7*). This confirms that the rhythmic activity and behavior can emerge from the modeled pre-motor circuitry itself, without a rhythmic input.

We also added 13B nodes, as shown in *Figure 6A*. These nodes receive inputs from the same 'black box' as the 13As.

*Figure 6B* shows three types of adjacency matrices obtained from the connectome. The adjacency matrices of the model are scaled versions of these three matrices. (We assume that the ratio between the weights in each adjacency matrix, rather than the absolute numeric values in *Figure 6B*, reflects the connectivity.) Each adjacency matrix is scaled separately (see Materials and methods,

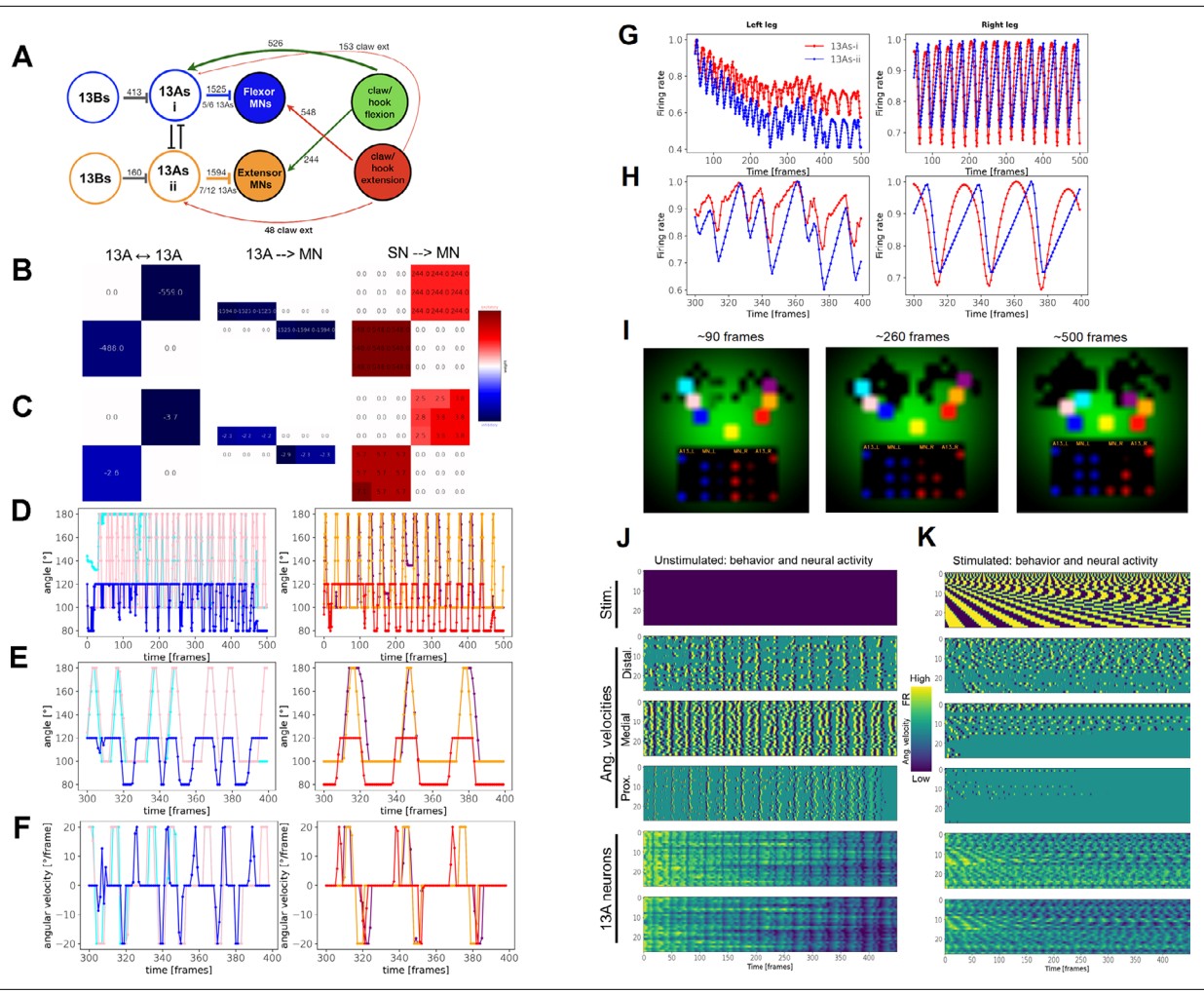

**Figure 6.** Modeling the 13 A circuits. Circuit diagram showing inhibitory circuits and synaptic weights based on connectome. Adjacency matrices from the empirically estimated weights, indicated in the simplified circuit diagram in (**A**). The 13B neurons in this model do not connect to each other, receive excitatory input from the black box, and only project to the 13As (inhibitory). Their weight matrix, with only two values, is not shown. Excitatory and inhibitory connections are shown in red and blue, respectively. Adjacency matrices of the model circuits are the same as in (**B**) but after fine-tuning. The three 'joint' angles of the left leg (left) and the right leg (right) as they change over the time of one episode (500 frames). Colors indicate 'joints' as follows: distal (cyan), medial (pink), and proximal (blue) for the left leg. Right leg: distal (purple), medial (orange), and proximal (red). The three 'joint' angles of the left leg (left) and the right leg (right) as they change over the time of one episode (500 frames). Colors indicate 'joint' angles in the same order as in (**I**). Same as (**D**) but zoomed-in to between 300 and 400 frames. Same as (**E**) but showing angular velocities [°/frame]. Firing rates (activity levels) of the two 13 A neurons (red and blue) over one episode (500 frames), for both legs (left, right). Same as (**G**) but zoomed-in to between 300 and 400 frames. Video frames from the beginning, middle, and end of a video of one episode. The left leg is represented by three 'joints': distal (cyan), medial (pink), and proximal (blue). Right leg: distal (purple), medial (orange), and proximal (red). The legs originate from the 'base' (yellow). As legs move over the 'body' (the environment – dust is represented as the green Gaussian distribution), the dust (green) is getting removed (black background). The bottom of each movie frame shows the activity of the two left 13 A nodes and six left MNs (blue). The right leg nodes are shown in red, on the right side. Brightness of the nodes indicates the activity level. See *Figure 6—video 1*. The dynamics of angular velocities of the left leg's 'joints', and left 13 A activation levels, over 100 episodes (500 frames each), when no stimulus is given (indicated by empty matrix on the top). Each row of each matrix is one episode. (The simulation started running for 50 frames before Time = 0 but it starts with very high peaks which were not plotted here for better visualization.). (**A**) Same as in J, but stimulation with pulses of varying durations is given. Top row of each matrix: pulse duration = 2 frames; bottom (100th) row of each matrix: pulse duration = 100 frames. The pulse stimulation is indicated in the top matrix.

The online version of this article includes the following video and figure supplement(s) for figure 6:

**Figure supplement 1.** Modeling the 13 A circuits.

**Figure 6—video 1.** Modeling the 13 A circuits.

https://elifesciences.org/articles/106446/figures#fig6video1

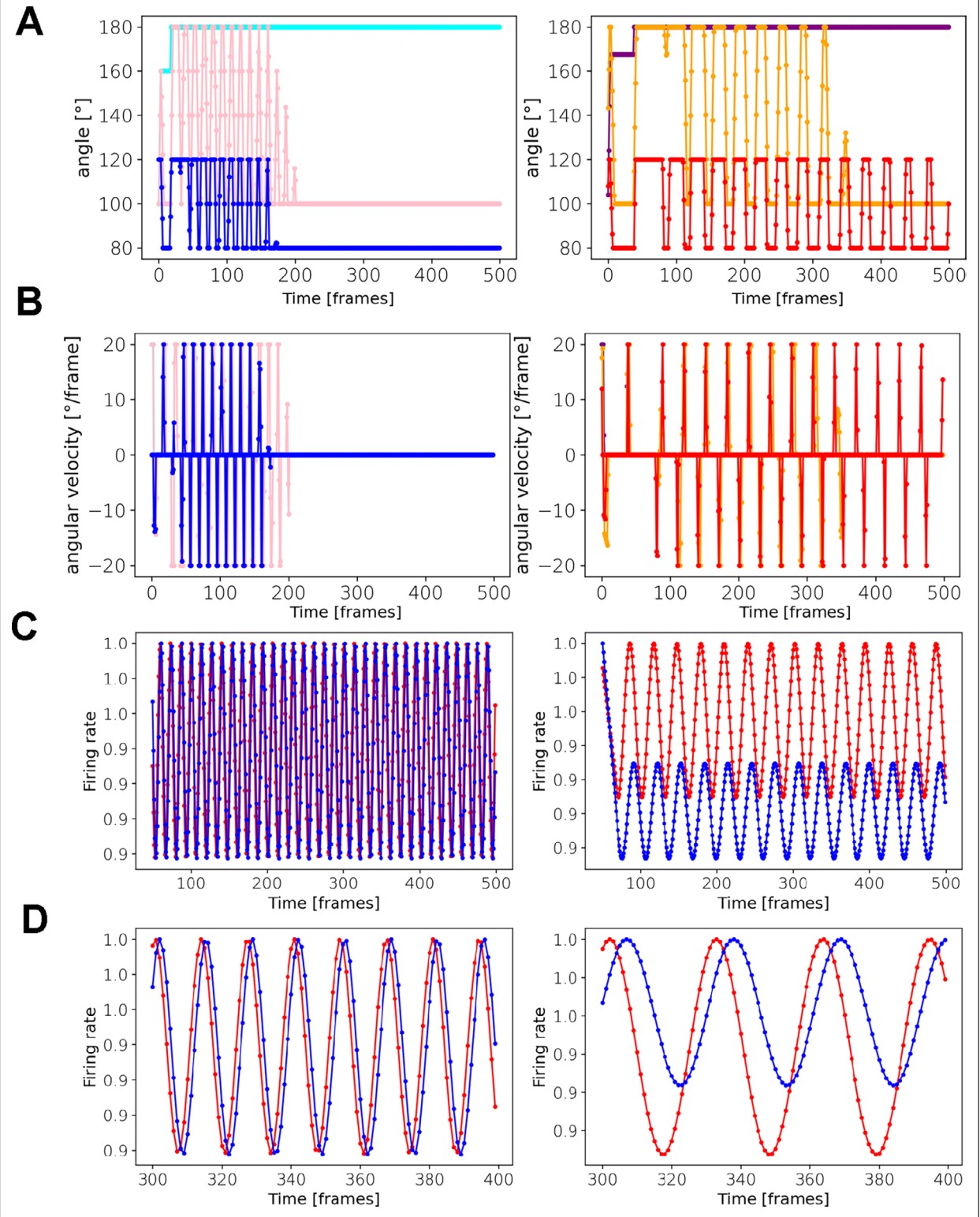

**Figure 7.** The modeled 13 A circuits can produce rhythmic behavior and activity without rhythmic external input. The three 'joint' angles of the left leg (left) and the right leg (right) as they change over the time of one episode (500 frames). Colors indicate 'joints' as follows: distal (cyan), medial (pink), and proximal (blue). Right leg: distal (purple), medial (orange), and proximal (red). Same as (**A**) but showing angular velocities [°/frame]. Firing rates (activity levels) of the two 13 A neurons (red and blue) over one episode (500 frames), for both legs (left, right). (**A**) Same as (**C**) but zoomed-in to between 300 and 400 frames.

the fine-tuning of the synaptic weights section), preserving the ratios of synaptic weights between the same types of neurons. When we run the model with the exact values of the adjacency matrices, we do not get any behavior. So, we instead allowed the model's parameters to deviate around their empirically obtained values. We can do this because our empirically derived weights are not exactly represented by the weights of the modeled network. Thus, we 'fine-tune' the weights, subject to constraints based on empirical values: the signs of the fine-tuned weights must remain the same as the empirical synapses (e.g. inhibitory neurons remain inhibitory), and their magnitudes have the upper and lower bounds of 20% above and below the empirical weights. No synapse can be removed from, or added to, the model by the fine-tuning process. The fine-tuning procedure was originally accomplished by genetic algorithms (GA) library PyGAD (https://pygad.readthedocs.io/en/latest/index.html), and is currently done by our own genetic algorithm (https://github.com/PrimozRavbar/Inhibitory-circuits copy archived at *Ravbar, 2025*). The fitness function is defined as the total amount of virtual dust that the model agent removes across several episodes of grooming. The genomes contain the parameters (synaptic weights and thresholds) and hyper-parameters (see Materials and methods). The fine-tuned weight matrices and the original ones are shown in *Figure 6B–C*. Note how the ratios between synaptic weights are largely preserved.

After the fine-tuning, we analyzed the activity of the modeled 13 A neural circuits and the behavior it produced. *Figure 6I* shows the first, middle, and last frame of a movie. The agent succeeded in removing most of the dust (the green pixels). *Figure 6D–F* shows the angles and angular velocities of the three 'joints' of each front leg, and *Figure 6G and H* show the corresponding neural activity of the 13 A neurons. Notice the periodic patterns in both the motor output and the firing rates of the 13 A nodes.

Next, we inquired how the model responds to perturbations analogous to the experimental activations of the 13 A neurons. *Figure 6J* shows the dynamics of left leg joint angles in 100 renditions, when no stimulus is applied. Notice the regular periodicity of these dynamics. As we vary the length of activation pulses (*Figure 6K*), behavior, as reflected in angle dynamics, becomes distorted. (Legs also lose coordination and consequently less dust is removed). These distortions also involve higher frequency of movements (see angular velocities in *Figure 6K*).

We also tried removing individual synaptic connections: removal of either one of the reciprocally inhibiting connections between 13As of a leg completely paralyzes it. Removing all 'proprioceptive' feedback from SNs to MNs does not stop the execution of the periodic movement, but it slows it down (*Figure 6—figure supplement 1*). Obliterating all 13 A → MN synapses, not surprisingly, completely paralyzes the leg. And, when we remove just the 13A-i-MN connections (which control the flexors of the right leg), we likewise get a complete paralysis of the leg. However, the removal of the 13A-ii-MN (which controls the extensors of the right leg) has only a modest effect on leg movements. So, we need the 13A-i neurons to inhibit the flexors (via motor neurons), but not extensors, in order to obtain rhythmic movements. Thus, our computational model confirms that rhythmic movements could be produced by inhibitory 13 A circuits (even without external sensory inputs from the dust, or patterned descending input). By replicating leg movements based on real anatomical connectivity, we investigated the potential functional roles of specific circuit components. This platform enhances our understanding of inhibitory circuits in leg coordination during grooming, providing a foundation for generating informed hypotheses in future experimental studies.

## Discussion
### Inhibitory circuit motifs
Using VNC EM connectome data (*Azevedo et al., 2024*; *Phelps et al., 2021*), we identified various circuit motifs formed by 13 A/ B inhibitory neurons that contribute to motor control.

### Feed-forward inhibition
Generalist 13 A neurons synapse onto multiple MNs influencing broad movements like whole leg extension, while specialist 13 A neurons could refine joint-specific movements (*Figure 2—figure supplement 1*). In stick insects, pilocarpine-induced rhythmic activity in deafferented nerve cords reveals strictly alternating activity between antagonistic motor neurons, and intracellular recordings show cyclic hyperpolarizing inputs to flexor and extensor motor neurons in antiphase (*Büschges,*

*1998*; *Büschges, 1995*). Similarly, locust thoracic ganglia exposed to pilocarpine exhibit alternating phases of antagonistic motor activity, indicating that inhibitory circuits are a conserved solution for generating alternating motor patterns (*Ryckebusch and Laurent, 1993*).

## Disinhibition

13 A neurons targeting extensor MNs connect to 13As targeting flexor MNs (*Figure 3*), enabling flexor activation when extensors are inhibited. Generalist-mediated disinhibition coordinates muscle synergies across joints, promoting leg extension or flexion. Some 13B neurons provide direct inhibition (*Figure 3A*), while others have an indirect effect by disinhibition of motor pools. This disinhibition motif, similar to those observed in motor systems for sequence selection (*Jovanic et al., 2016*; *Chevalier and Deniau, 1990*; *Mink, 1996*; *Grillner et al., 2005*; *Benjamin et al., 2010*; *Zhu et al., 2024*; *Zhao et al., 2019*) and flight regulation (*Zhao et al., 2019*), may prime motor responses by holding them at the ready, to be released when inhibition is removed. Moreover, alternating inhibition and disinhibition of antagonistic motor neuron pools by premotor CPG networks has been observed previously in stick insects (*Büschges, 1998*; *Büschges, 2005*; *Ruthe et al., 2024*). Disinhibitory motifs are also present in *Drosophila* larvae: interlinked inhibitory interneurons implement lateral and feedback disinhibition to guide sensorimotor decisions (*Jovanic et al., 2016*), and selective inhibition by premotor interneurons coordinates the sequential activation of motor neurons (*Zwart et al., 2016*). Our connectome analysis extends these concepts by identifying inhibitory 13 A/B motifs that could implement such coordination in the adult *Drosophila* VNC to control leg movements.

## Reciprocal inhibition

13 A neurons inhibiting flexors and extensors within a leg are reciprocally connected (*Figure 3*). This circuit resembles reciprocal inhibitory motifs observed in vertebrate locomotor circuits, such as rIa interneurons (*Kiehn, 2016*; *Talpalar et al., 2011*; *Cowley and Schmidt, 1995*; *Kjaerulff and Kiehn, 1997*; *Whelan et al., 2000*; *Endo and Kiehn, 2008*; *Zhang et al., 2014*) as well as in invertebrate CPGs, including the swimming network of *Clione limacina* and the heartbeat CPG of the leech (*Satterlie, 1985*; *Arshavsky et al., 1998*; *Cymbalyuk et al., 2002*; *Weaver et al., 2010*), highlighting a conserved solution for generating alternating motor activity.

Our connectomic analysis extends these classic findings by demonstrating reciprocal inhibition between inhibitory premotor 13 A neurons themselves, rather than solely between antagonistic motor neurons. This additional layer of reciprocal connectivity suggests that 13As may act as a rhythm-generating kernel that shapes the timing of motor output and coordinates antagonistic activation.

## Redundant inhibition

Inhibitory neurons target both MNs and their excitatory pre-synaptic partners, creating a parallel pathway that could modulate motor output through direct and indirect inhibition. For example, 13A-10f-α connects to both tibial flexor MNs and excitatory premotor neurons (20 A/3 A) that activate flexor MNs (*Figure 2—figure supplement 4*), preventing their activation by two parallel pathways. Similarly, the 13B-4i neuron connects to 13 A neurons inhibiting flexor MNs, leading to disinhibition (*Figure 3—figure supplement 2*), and to excitatory neurons presynaptic to 13 A neurons, thereby both removing inhibition from flexor MNs and limiting the excitatory drive onto 13 A neurons. This motif has also been described in the locust flight central pattern generator, where inhibitory interneurons target both motor neurons and their excitatory premotor inputs (*Robertson and Pearson, 1985*).

## Proprioceptive feedback onto inhibitory circuits

Position-sensing proprioceptors connect to 13As (*Figure 3D*, *Figure 3—figure supplement 3*), which inhibit flexion and disinhibit extension and vice versa, complementing reciprocal inhibition to generate alternation. Movement-sensing proprioceptors synapse onto 13 A neurons, but are suppressed during walking and grooming (*Dallmann et al., 2024*). Sherrington's 1910 proposal, supported by spinal cat studies, suggests proprioception triggers alternation—a mechanism observed in rhythmic behaviors like locust flight and mammalian respiration (*Sherrington, 1910*; *Andersson and Grillner, 1983*; *Field and Burrows, 1982*; *Kriellaars et al., 1994*; *Pearson, 2004*; *Marlot et al., 1987*; *Jammes et al., 1986*). Connections between position sensors, 13 A neurons, and antagonistic MNs suggest that proprioceptive signals may trigger alternation of leg movements during grooming.

Local and descending neurons presynaptic to reciprocally connected 13 A neurons could also induce alternation. The balance between internal circuits within the central nervous system and sensory feedback contributes to pattern generation. Future studies will dissect the extent of peripheral vs. central control in generating alternation.

These motifs can explain the spatial and temporal dynamics of grooming movements. *Flexors and extensors at multiple joints must coordinate to fully extend or contract a leg.* Neurons targeting related MNs could facilitate synchronization. For example, cluster 6 or 8 13 A neurons target proximal and medial joint extensor MNs, allowing leg flexion. During head sweeps, proximal and medial joints move independently—for example, the proximal joint flexes while the medial joint extends—which can be coordinated by cluster 10 13 A neurons and 13A-9e, –9 f.

## Flexors and extensors should be mutually exclusive

A generalist connected to both proximal and medial joint MNs could facilitate leg rubbing. For example, 13 A-10c synchronously inhibits proximal, medial, and distal flexor MNs (*Figure 3C*, *Figure 2—figure supplement 1I4*) while targeting 13 A neurons (13 A-8a, –8b, and 8 c) connected to extensor MNs (*Figure 3F*, *Figure 2—figure supplement 1G1-G3*). This arrangement ensures inhibition of MNs and disinhibition of antagonistic MNs across multiple joints, preventing simultaneous coactivation.

### Flexors and extensors alternate or co-contract

Reciprocal inhibition and proprioceptive feedback onto 13 A neurons could facilitate alternation between extension and flexion. Reciprocal inhibition between generalist neurons of 13As-i and 13As-ii could induce alternation during head sweeps (*Figure 3D*). This aligns with the half-center model, which proposes that rhythmic motor patterns arise from two mutually inhibitory neuronal populations that alternate their activity in an out-of-phase manner to drive opposing muscle groups (*Brown, 1911*; *Brown, 1914*; *Stuart and Hultborn, 2008*). Our behavior experiments and modeling further support this connectivity, as we indeed induced rhythmic motion through pulsed activation of specific inhibitory neurons without altering any excitatory drive. However, flexor–extensor co-contraction can also be functionally relevant, such as for modulating joint stiffness during postural stabilization or for generating large forces required for fast movements (*Zakotnik et al., 2006*; *Günzel et al., 2022*; *Ogawa and Yamawaki, 2025*). Some generalist 13 A neurons could facilitate co-contraction across different leg segments, but none target antagonistic motor neurons controlling the same joint. Therefore, co-contraction within a single joint would require the simultaneous activation of multiple 13 A neurons, potentially coordinated by upstream neurons that can recruit multiple 13As.

## Inhibitory innervation imbalance between flexors and extensors

We observed an imbalance in the innervation: more 13 A neurons target extensors than flexors across multiple joints. While legs alternate between extension and flexion, they remain elevated during grooming. To maintain this posture, some MNs must be continuously activated while their antagonists are inactivated. Uneven distribution of inhibition could ensure that while some MNs remain active, others alternate flexion and extension in a controlled manner. Among the 13 A neurons inhibiting antagonistic muscles, reciprocally connected ones could induce alternation, while others could keep legs elevated.

The asymmetry in connections, with more 13B neurons disinhibiting flexor-inhibiting 13As, suggests a mechanism for preferential flexion, supporting the flexor burst generator model, where the generator actively excites flexor MNs while inhibiting tonically active extensors (*Duysens et al., 2013*; *Zhong et al., 2012*).

It is important to note that our interpretations are based on connectome-derived connectivity patterns and behavioral observations from optogenetic manipulations, without direct physiological recordings from motor neurons or muscles. Studies in locust, stick insect, and cockroaches have demonstrated that fast cyclic leg movements can emerge from the interaction of rhythmic activity in a single muscle with the passive tension of its antagonist (*Brown, 1911*; *Heitler, 1974*; *Bennet-Clark, 1975*; *Page et al., 2008*; *Hooper et al., 2009*). Therefore, while our connectome analysis suggests motifs for multi-joint coordination, the exact patterns of motor neuron and muscle activity during grooming remain to be empirically confirmed.

## Grooming command neuron pathways to 13A neurons

Descending neurons that command grooming movements, such as aDN (*Hampel et al., 2015*) and DNg12 (*Guo et al., 2022*), synapse onto 13 A neurons. Although it remains unclear how constitutive activity in these descending neurons generates rhythmic grooming, the reciprocal inhibition circuits among 13As could provide a mechanism. The antennal grooming command neuron aDN synapses onto two primary 13As (γ and α; 13As-i) that connect to proximal extensor and medial flexor motor neurons, as well as four other 13As (9 a, 9 c, 9i, 6e) projecting to body wall extensor motor neurons. These 13As-i also form reciprocal connections with 13As-ii, potentially supporting oscillatory leg movements. aDN connects to homologous 13As on both sides, consistent with the bilateral coordination required for antennal sweeping.

The head grooming/leg rubbing command neuron DNg12 synapses onto ~50 13 As, primarily those projecting to proximal motor neurons.

While structural connectivity highlights candidate pathways for rhythmic movement generation, the dense interconnections among command neurons and premotor circuits suggest that multiple motifs likely contribute to the observed behaviors. Further work is needed to determine how these pathways are dynamically recruited during natural grooming sequences.

## Spatial mapping of premotor neurons in the nerve cord

The organization of 13 A neurons—where morphology, position, and motor-neuron connectivity align—suggests a premotor topographic map in the fly VNC. While 13 A neurons partially reflect the myotopic map of MN dendrites (*Baek and Mann, 2009*; *Brierley et al., 2012*; *Landgraf et al., 2003*), they do not target single MNs. Instead, they form clusters that connect either to a few MNs within a segment (specialists) or to groups spanning multiple leg segments and joints (generalists). Broadly projecting generalists could recruit or inhibit coordinated sets of muscles, effectively encoding multi-joint movement patterns, whereas specialists provide more precise, segment-specific control. In other words, this map represents both the leg-muscle architecture and premotor synergies—spatially organized modules for complex actions.

Such spatial logic is consistent with other mapped systems: sensory maps in the antennal lobe preserve receptor identity (*Vosshall et al., 2000*; *Wong et al., 2002*; *Gao et al., 2000*), proprioceptors form an orderly representation of joint angles and forces in the VNC (*Mamiya et al., 2023*), or vertebrate spinal cords display dorsoventral recruitment gradients that scale locomotor output (*McLean et al., 2007*; *Gatto et al., 2021*). Together, these findings suggest that the fly VNC contains a layered spatial organization—from proprioceptive inputs to premotor and motor outputs—that could allow descending and sensory pathways to flexibly engage movement modules and synergies by addressing specific regions of neuropil. This raises the possibility that modular movement primitives encoded in the VNC can be combined to generate diverse leg behaviors, including grooming and walking.

## Computational modeling of inhibitory circuits

We employ a linear rate-based (non-spiking) neural network to represent the 13 A circuitry. Natural behavior and neural activity are, of course, more complex. Our model explores how 13 A circuit motifs could contribute to, or produce de novo, behavioral features, including rhythmic movements (which might be otherwise produced by upstream circuits), in 2D space, matching the dimensionality of our behavioral data. This abstraction is inspired by *Jovanic et al., 2016*; *Büschges, 1995*, but we add the agent component to the neural circuitry and apply evolutionary 'fine-tuning' of parameters. Similar circuit motifs, namely reciprocal inhibitions between pre-motor neurons and the sensory feedback, have been modeled before, in particular neuroWalknet, where such motifs do not require a separate CPG component to generate rhythmic behavior (*Schilling and Cruse, 2020*; *Schilling and Cruse, 2023*). However, our neuronal model is much simpler than the neuroWalknet - it controls a 2D agent operating on an abstract environment (the dust distribution). In real animals or complex mechanical models such as NeuroMechFly (*Lobato-Rios et al., 2022*; *Wang-Chen et al., 2024*), a more explicit central rhythm generation may be advantageous for the coordination across more degrees of freedom. When we perturb the model by activating 13 A neurons with varying pulse lengths, we observe decreased coordination and increased movement frequency (compare angular velocities in *Figure 6J* with *Figure 6K*). Longer pulses can almost completely paralyze a leg (*Figure 6K*). Here,

we did not attempt to simulate the exact experimental procedures. In the future, the model's parameters could be fine-tuned within similarly constrained space, but with the fitness function modified: instead of parameters being optimized to remove 'dust', they could be optimized by the similarity of behavioral features between the model and real flies, as has been done in whole-animal modeling (*Karashchuk et al., 2025*), under various experimental conditions.

### Future directions

Our work lays groundwork for future exploration of the functional contributions of inhibitory circuits to motor control. Developing genetic tools to target specific inhibitory neurons and functional imaging during behavior would allow us to correlate temporal neuron activation with limb motion. While we focus on inhibitory control of one leg, manipulating these neurons reveals defects in left-right coordination. Investigating the circuitry involved—possibly mediated by commissural and/or descending

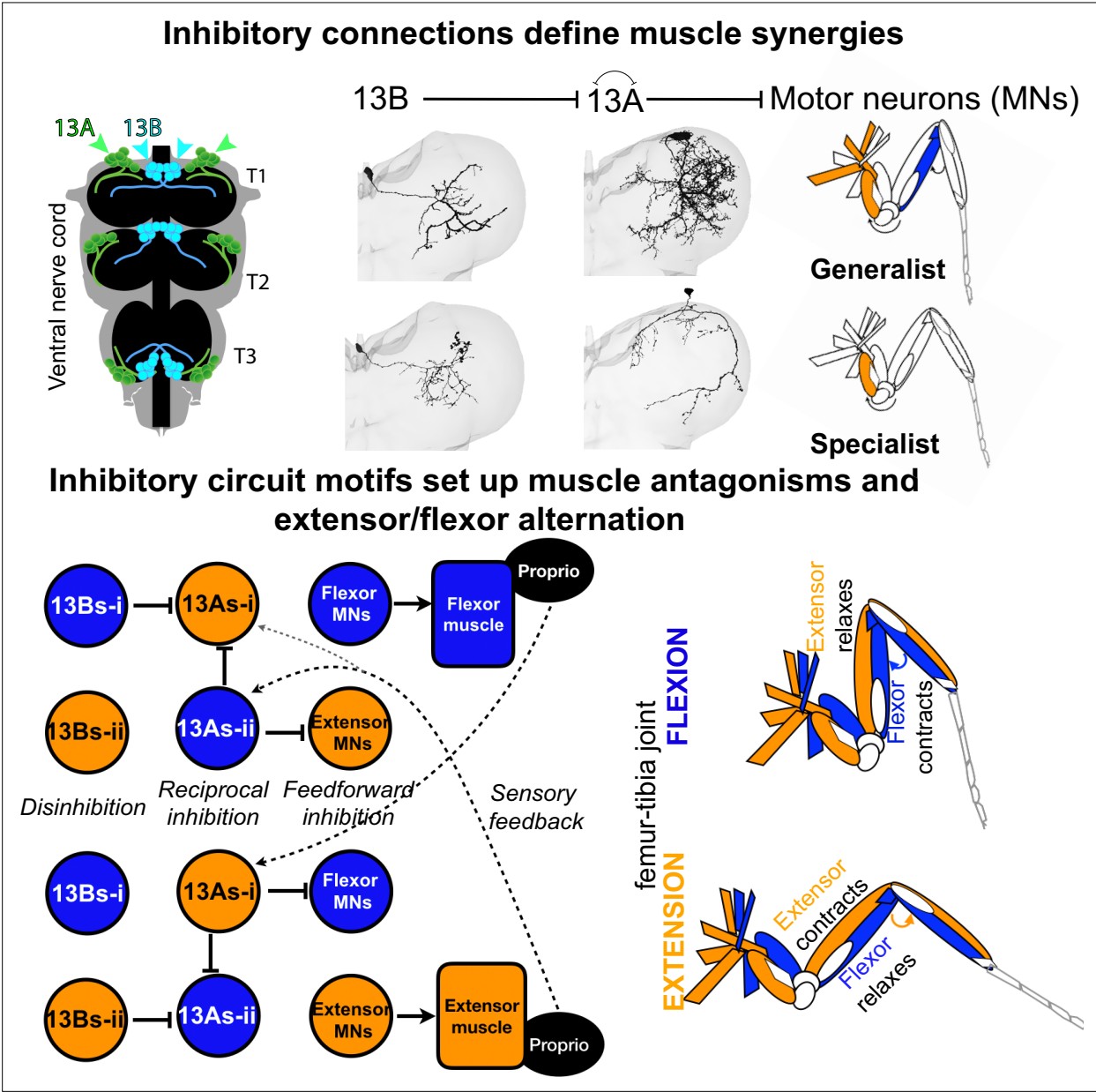

**Figure 8.** Summary of inhibitory neuron contributions to leg movement coordination. Inhibitory neurons from the 13A and 13B hemilineages in the Ventral Nerve Cord connect to combinations of leg motor neurons, coordinating groups of muscles to work together in synergy. Circuit motifs connecting inhibitory neurons may support muscle antagonisms and extensor/flexor alternation.

neurons connected to these circuits—remains to be explored to elucidate the underlying circuits. Pulsed activation of 13 A neurons induces grooming or walking in clean flies, while manipulating their activity in dusted flies alters grooming timing. This suggests that 13 A neurons may be part of central pattern generators. While our experiments with multiple genetic lines labeling 13 A/B neurons consistently implicate these cells in leg coordination, ectopic expression in some lines raises the possibility that other neurons may also contribute to these phenotypes. In addition, other excitatory and inhibitory neural circuits, not yet identified, may also contribute to the generation of rhythmic leg movements. Future studies should identify such neurons that regulate rhythmic timing and their interactions with inhibitory circuits.

Our connectome analysis reveals that inhibitory neurons within a hemilineage form circuit motifs with complex connections to specific leg motor neurons, as summarized in *Figure 8*. These neurons likely complement excitatory premotor circuits, enabling multifunctionality in behaviors such as grooming and walking. Their connectivity suggests roles in coordinating movements across multiple joints, enforcing flexor/extensor muscle antagonism, and driving extension and flexion alternation. We model and extrapolate potential functions of these complex inhibitory motifs. Normal activity of these inhibitory neurons is important for grooming; silencing or continuously activating them reduces time spent and effectiveness in dust removal, leading to forced flexion or abnormal extension, with limbs locked in extreme poses. Moreover, we show that inhibition, independent of excitatory input, plays an instructive role in generating rhythmic leg movements. Although we focus on grooming behavior, we expect these motifs to also contribute to walking, as the leg must recruit limited motor and premotor components to generate diverse movements. We conclude that inhibitory neurons are essential for controlling flexible limb movements and may play a key instructive role in timing and coordinating rhythmic behaviors.

## Materials and methods

**Key resources table**

| Reagent type (species) or resource | Designation | Source or reference | Identifiers | Additional information |
|---|---|---|---|---|
| Antibody | Chicken polyclonal anti-GFP | Abcam | RRID:AB_300798 | (1:1000) |
| Antibody | Rabbit (Rb) polyclonal anti- GFP | Invitrogen | Cat #A11122; RRID:AB_221569 | (1:1000) |
| Antibody | Mouse (ms) monoclonal anti-Bruchpilot | DSHB | RRID:AB_2314866 | (1:200) |
| Antibody | ms anti-Neuroglian (BP104) | DSHB | RRID:AB_528402 | (1:40) |
| Antibody | Mouse polyclonal anti-V5:DyLight 550 | AbD Serotec | RRID:AB_2687576 | (1:300) |
| Antibody | Rabbit polyclonal anti-HA | Cell Signaling Technologies | RRID:AB_1549585 | (1:300) |
| Antibody | Rat monoclonal anti-FLAG | Novus Bio | RRID:AB_1625982 | (1:200) |
| Antibody | Goat anti-Chicken Alexa Fluor 488 | Invitrogen | RRID:AB_142924 | (1:400) |
| Antibody | Goat anti-rabbit Alexa Fluor 488 | Invitrogen | RRID:AB_143165 | (1:400) |
| Antibody | Goat anti-mouse Alexa Fluor 568 | Invitrogen | RRID:AB_2534072 | (1:400) |
| Antibody | Goat anti-mouse Alexa Fluor 633 | Invitrogen | RRID:AB_2535719 | (1:400) |
| Antibody | Goat anti-rabbit Alexa Fluor 568 | Invitrogen | RRID:AB_143157 | (1:400) |
| Antibody | Goat anti-rat Alexa Fluor 488 | Invitrogen | RRID:AB_2534074 | (1:400) |
| Antibody | Donkey anti-rat Alexa 647 | Jackson ImmunoResearch | RRID:AB_2340694 | (1:400) |
| Chemical compound | Insect-a-slip | Bio Quip Products | Cat#2871 A | |
| Chemical compound | Reactive Yellow 86 | Organic Dyestuffs Corporation | CAS 61951-86-8 | |

*Continued on next page*

*Continued*

| Reagent type (species) or resource | Designation | Source or reference | Identifiers | Additional information |
|---|---|---|---|---|
| Genetic reagent (*Drosophila melanogaster*) | R35G04-GAL4-DBD | Bloomington Stock Center (BDSC) | RRID:BDSC_70351 | |
| Genetic reagent (*Drosophila melanogaster*) | GAD-GAL4-AD | *Diao et al., 2015* | | |
| Genetic reagent (*Drosophila melanogaster*) | GAD-GAL4-DBD | Gift from Haluk Lacin and James Truman | | |
| Genetic reagent (*Drosophila melanogaster*) | Dbx-GAL4-AD | Gift from Haluk Lacin and James Truman | | |
| Genetic reagent (*Drosophila melanogaster*) | Dbx-GAL4-DBD | Gift from Haluk Lacin and James Truman | | |
| Genetic reagent (*Drosophila melanogaster*) | R11C07 AD | BDSC | RRID:BDSC_70533 | |
| Genetic reagent (*Drosophila melanogaster*) | w[1118]; P{y[+t7.7] w[+mC]=20XUAS-IVS-CsChrimson.mVenus}attP40 | BDSC | RRID:BDSC_55135 | |
| Genetic reagent (*Drosophila melanogaster*) | P{JFRC7-20XUAS-IVS-mCD8::GFP}attp40 | BDSC | RRID:BDSC_32194 | |
| Genetic reagent (*Drosophila melanogaster*) | UAS-GTACR1 | Gift from Adam Claridge-Chang | | |
| Genetic reagent (*Drosophila melanogaster*) | w[*]; P{y[+t7.7] w[+mC]=UAS-GtACR1.d.EYFP}attP2 | BDSC | RRID:BDSC_92983 | |
| Genetic reagent (*Drosophila melanogaster*) | "w[1118] P{y[+t7.7] w[+mC]=R57C10-FLPL}su(Hw)attP8; PBac{y[+mDint2] w[+mC]=10xUAS(FRT.stop)myr::smGdP-HA}VK00005 P{y[+t7.7] w[+mC]=10xUAS(FRT.stop)myr:: smGdP-V5-THS –10xUAS(FRT.stop)myr::smGdP-FLAG}su(Hw) attP1"(MCFO3)" | BDSC | RRID:BDSC_64087 | |
| Genetic reagent (*Drosophila melanogaster*) | 10XUAS-IVS-eGFP-Kir2.1 | *von Reyn et al., 2017* | | |
| Genetic reagent (*Drosophila melanogaster*) | "Control-GAL4-AD-GAL4-DBD EMPTY SPLIT: BPp65ADZp(attp40); BPZpGDBD(attp2)" | BDSC | RRID:BDSC_79603 | |
| Software, algorithm | DeepLabCut | *Mathis et al., 2018* | RRID:SCR_021391 | |
| Software, algorithm | Python | | RRID:SCR_008394 | |
| Software, algorithm | MATLAB | MathWorks | RRID:SCR_001622 | |
| Software, algorithm | FIJI | *Schindelin et al., 2012* | RRID:SCR_002285 | |
| Software, algorithm | Adobe illustrator | | RRID:SCR_010279 | |
| Software, algorithm | Adobe Photoshop | | RRID:SCR_014199 | |
| Software, algorithm | Braincircuits | | https://braincircuits.io/ app?p=fruitfly_fanc_ public | |

*Continued on next page*

*Continued*

| Reagent type (species) or resource | Designation | Source or reference | Identifiers | Additional information |
|---|---|---|---|---|
| Software, algorithm | Neuroglancer | *Maitin-Shepard et al., 2021* | RRID:SCR_015631 | |
| Software, algorithm | fancr | *Azevedo et al., 2024*; *Jefferis, 2024* | https://github.com/flyconnectome/fancr | |
| Software, algorithm | neuPrint | *Plaza et al., 2022* | https://neuprint.janelia.org/ | |
| Software, algorithm | CATMAID | *Saalfeld et al., 2009* | RRID:SCR_006278 | |
| Software, algorithm | RStudio | | RRID:SCR_000432 | |

## Contact for reagent and resource sharing

For information and inquiries regarding resources and reagents, please write to the lead contact Julie H. Simpson (jhsimpson@ucsb.edu).

## Experimental model and subject details

*Drosophila melanogaster* were raised on a standard cornmeal medium at 25 °C in a 12 hr light/dark cycle. For optogenetic experiments, one-day-old flies were transferred to food containing 0.4 mM all-trans-retinal and kept in the dark for 3 days. Genotypes of the fly lines are included in the Key Resources Table.

## Identification of fly lines that target inhibitory neurons

We visually screened the VNC expression of various GAL4 lines on Flylight database (*Jenett et al., 2012*) and compared them to the inhibitory hemilineages (*Harris et al., 2015*). Next, we obtained corresponding *AD and DBD* flies from BDSC for the candidate lines and crossed them with *GAD-GAL4-AD or GAD-DBD* (Haluk Lacin) to confirm and restrict their expression in GABAergic neurons. *R35G04-GAL4-DBD, GAD-GAL4-AD* labeled six 13 A and six 13B neurons per hemisegment. We also isolated six 13 A neurons from this line by using *R35G04-GAL4-DBD, DBX-GAL4-AD* combination. *R11C07-GAL4-DBD and GAD-GAL4-AD* labels 4 inhibitory neurons. We intersected *R11C07 DBD* with *Dbx AD* to isolate two 13 A neurons. *R11B07-GAL4-DBD, GAD-GAL4-AD* labeled 3 13B neurons.

## Immunofluorescence and confocal microscopy

Flies were immobilized by anesthetizing them on ice (4 °C). The central nervous system (CNS) was carefully dissected in 1 X Phosphate-buffered saline (PBS). Subsequently, the wings were removed, and the flies were positioned ventral side up on a Sylgard plate. All legs were excised, and fine forceps (No. 5 Dumont from FST, Switzerland) were employed to delicately open the thorax along the midline, taking care to avoid damaging the underlying thoracic ganglia. A small incision near the first abdominal segment ensured preservation of the abdominal ganglion. Surrounding tissues were cleared from the thoracic ganglia, which were then gently extracted by grasping the neck connective. The dissected thoracic ganglia were subsequently fixed in 4% buffered paraformaldehyde for 45 min at 4 °C.

Post-fixation, the thoracic ganglia underwent three 15-min washes in 0.1% Triton X-100 (PBT) at room temperature on a shaker at 60 rpm, followed by a 20-min wash in 0.1% PBT with normal goat serum (blocking solution). Primary antibodies, diluted in 0.1% PBT-NGS, were applied to the samples and incubated overnight at 4 °C on a horizontal shaker. Following primary antibody incubation, the samples underwent three 15-min washes in 0.1% PBT and one 20-min wash in 0.1% PBT-NGS. Secondary antibodies, diluted in 0.1% PBT-NGS, were added to the samples and incubated for 2–4 hr at room temperature on a shaker at 60 rpm. Secondary antibody removal was achieved through four 15-min washes with 0.1% PBT at room temperature. Finally, the tissues were mounted on glass slides using Vectashield mounting medium (Vector Labs).

Primary antibodies used were Chicken pAb anti-GFP (Abcam, 1:1000), Rabbit (Rb) anti- GFP (Abcam, 1:1000), mouse (ms) anti-Neuroglian (BP104) (DSHB, 1:40), ms monoclonal anti-Brp antibody

(nC82) (DSHB, 1:200). For MCFO labeling experiments (*Nern et al., 2015*), Rb mAb anti-HA (Cell Signaling Technologies, 1:300), Rat anti-FLAG (Novus Biologicals, 1:200), DyLight549-conjugated anti-V5 (AbD Serotec; 1:300 dilution).

Secondary antibodies from Invitrogen Molecular Probes conjugated with Alexa-488, Alexa-568, and Alexa-647 raised against chicken, ms, and Rb were used in 1:400 dilution.

Zeiss LSM710 confocal microscope was used to obtain images of the CNS. Images were then processed in FIJI.

## Recording and analysis of grooming in clean and dusted flies

For open field assay, we dusted the flies and obtained the recording as previously described (*Schilling and Cruse, 2020*). Constant light intensity of 5.6 mW/cm² was used for continuous activation. Automated behavior analysis (ABRS) was used to quantify the amount of time flies perform individual grooming actions (*Ravbar et al., 2019*). Additionally, manual scoring was performed by *Berkowitz and Laurent, 1996a* in flies showing uncoordinated leg movements. Quantification and statistical analysis describing the percentage of time dusted flies spent doing grooming and uncoordinated leg movements upon 13 A and B activity manipulation was performed in Matlab as previously described (*Zhang et al., 2020*).

For limb tracking, either clean or dusted freely moving male flies were put in a studio containing 10 mm diameter quartz chamber and 100 Hz videos were recorded from below using FLIR Blackfly S camera. Custom-built LED panels (LXM2-PD01-0050, 625 nm) were utilized to deliver light activation from below, with an intensity of 1.1 mW/cm². Green LED was used for silencing experiments.

## Behavior analysis

Raw data, consisting of coordinates of the six annotated points on the front legs and two reference points on the body (per one frame of a video), was obtained from DeepLabCut76. From these coordinates, we computed: (1) (fly-centric) spatial positions of the body parts, (2) the spatial velocities of the points, (3) the whole-body velocity (the translation, obtained from the reference points in absolute coordinates), (4) the Euclidean distances between the leg points ('joints') and other 'joints' or the reference points, and the joint angles.

All behavior analysis was performed using Python, version 3.9.7.

### Continuous feature extraction

The Euclidean distances between various body parts (the six 'joints' and the two reference points) are used as continuous features. The body velocity was computed as a Euclidean distance of a point covered across a 50 frame (0.5 s) time window. Three joint angles per leg were computed from the three points on the leg and the two reference points. These angles are: posterior, medial, and distal angles. Angular velocities were computed as derivatives of the raw angles and were lightly smoothed by a Gaussian filter (filter sigma = 2 frames). The Euclidean distances were also smoothed by the same method.

The angular velocity time series (AV) is used as the basic signal from which other continuous behavioral features are extracted, and also serves as the basis for segmentation. The main continuous features extracted from the AV are the phase and the frequencies.

The phase (between the movement of the front legs) was computed from cross-correlation of AV signals of the two legs, using the signal.correlation_lags() function with the window for cross-correlation of five frames. We took the time position of the maximum cross-correlation (the peak) as the lag (the phase). If the two legs are perfectly in phase, the peak of the cross-correlation will be at time 0. The phase is a good indicator of a general type of grooming behaviors: front-leg rubbing is usually associated with a non-zero phase (a lag), whereas the head-cleaning more often than not has a zero lag (legs moving in-phase).

Frequencies are computed from the AV signal as well. We use the numpy.fft.fft() function with the time window of 25 frames (0.25 s) and the input size of 160 frames (the length of the signal with the zero-padding) to compute the spectrum. The frequency at which the spectral power is maximal is taken as the frequency of the signal.

## Segmentation

Our analysis is focused on individual leg movements during grooming. These include contractions and extensions of the front-leg joints. For this purpose, we separate the continuous features into segments corresponding to these movements.

The segmentation is performed by applying a stationary threshold to the angular velocity (AV) of the proximal joint angles. These joints determine the whole leg movements (flexions and extensions), so we consider them suitable for the purpose of segmentation. Next, for each segment, we compute the averages, minima, and maxima of each continuous feature within that segment. We also compute the duration of each segment as an additional feature. These *segmented features* are used for behavioral classification and for further analysis, including comparing different groups of flies.

## Classification of behavior and dimensionality reduction

We collect segments (segment features) from data sets of groups of flies and pool them together for the purpose of classification. (e.g. experimental and control data sets are used together.) Altogether we use 63 segment features, including averages of: Euclidean distances, AVs, phases between joint pairs, frequencies of all joints, whole-body velocity, segment durations. We also include the maxima of the six AVs, and we can include combined features too (here we only use one combined feature – AV*segment duration).

For better mapping and classification of segments, we account for the temporal context of a segment: for each segment's set of features, we add the same type of features of the next two segments (in time). Thus, we analyze features of triplets of segments rather than single segments.

The feature matrix X of the size f x t, where t is the total number of segments and f is the number of segment features multiplied by 3 (to account for the temporal context), is the input to the UMAP dimensionality reduction algorithm.

We use UMAP from https://pypi.org/project/umap-learn/ with the following parameters:

    n_neighbors = 350,
    min_dist = 0.1,
    n_components = 2

The **X** is thus projected onto two dimensions. For the UMAP construction, we can remove some of the data from **X**. Namely, we can remove the data points (segments) where angular velocity of proximal joints is below a certain value (fly is presumably not moving) or when the quality of data is too low (see quality control).

For the classification, we apply the agglomerative clustering onto the 2D UMAP projection. We use AgglomerativeClustering() from sklearn.cluster (here). We select the number of clusters of 12–14. Thus, we produce 12–14 class labels that can be applied to the 63 features of the segment feature matrix **X**.

Now we can compare the different behavioral classes, across the 63 features, between various groups of flies.

## Comparing groups of flies

With multiple behavioral classes and features, we can now compare different experimental groups of flies to assess the behavioral effects of the experimental procedures. For each *class-feature pair*, we can determine whether there is a significant difference between the groups. Experimental and control groups contain data (segments) from multiple animals, so the segments from the same animal are not independent (clustered data). We therefore applied Linear Mixed Models (LMMs) to comparisons between the groups. This was accomplished by the Python method mixedlm() from statsmodels.formula.api library (https://www.statsmodels.org/stable/api.html).

The groups were compared across several features-class pairs. This calls for the multiple hypotheses adjustment. We applied the False Discovery Rate method, using the statsmodels.stats.multitest function (here). The alpha parameter was set to 0.05, and the method argument was set to fdr_bh' (see the code at https://github.com/SyedDurafshan/Inhibitory-circuits-Source; *Syed, 2026*).

## Model

### The 13A circuits

The model of 13 A and associated circuits was built from simple linear neuronal networks. We did not model individual neurons but rather abstracted them to nodes, interconnected by synaptic weights corresponding to the numbers of synapses obtained from the connectome analysis. Since the 13 A circuits in the right and left front leg neuropil are mirrored and there are slightly more connections on the left side (*Figure 2—figure supplement 2*), the synaptic weights of the same circuits on the left neuropil were used for modeling purposes.

The network is shown in *Figure 6A*. Below is the description of the 13 A circuit model for one leg. The full model consists of two (front) legs, built the same way.

The two 13 A nodes connect to each other reciprocally (inhibitory synapses). The adjacency matrix for these nodes is:

$$W_{13A\leftarrow\rightarrow 13A} = \begin{bmatrix} 0 & -559 \\ -488 & 0 \end{bmatrix}$$

The activation levels of the 13As are simply:

$$A^{t+1}_{13A} = A^t_{13A} + W^T_{13A\leftarrow\rightarrow 13A} * A^t_{13A} - \Theta_{13A}$$

**A**13A is a vector of activation levels of the 13As, and **θ**13A is a vector of thresholds of 13 A neurons.

The 13 A nodes inhibit the six motor neurons (MNs). These connections are represented by the adjacency matrix:

$$W_{13A\rightarrow MN} = \begin{bmatrix} -1594.0 & -1525.0 & -1525.0 & 0.0 & 0.0 & 0.0 \\ 0.0 & 0.0 & 0.0 & -1525.0 & -1594.0 & -1594.0 \end{bmatrix}$$

The six MNs act on the three pairs of 'antagonistic muscles', which in turn control the changes of joint angles (three per leg). Flexion and extension rates of the antagonistic muscles are directly proportional to the activation levels of the MNs.

The activation levels of the MNs are:

$$A^{t+1}_{MN} = A^t_{MN} + W^T_{13A\rightarrow MN} * A^t_{13A} - \Theta_{MN}$$

**A**MN is a vector of activation levels of the MNs, **A**13A is a vector of activation levels of 13As, and **θ**MN is a vector of thresholds of 13 A neurons.

The inputs to the six sensory neurons (SNs) are directly proportional to the flexion and extension rates of the joint muscles. So, the activation levels of the SNs are:

$$A^{t+1}_{SN} = A^t_{SN} + extension/flexionrate - \Theta_{SN}$$

where **A**SN is a vector of activation levels of SNs, and **θ**SN is a vector of thresholds of SNs.

The SNs then feed back onto both, the MNs and the 13As. The adjacency matrix representing the sensory feedback to the MNs is:

$$W_{SN\rightarrow MN} = \begin{bmatrix} 0.0 & 0.0 & 0.0 & 244.0 & 244.0 & 244.0 \\ 0.0 & 0.0 & 0.0 & 244.0 & 244.0 & 244.0 \\ 0.0 & 0.0 & 0.0 & 244.0 & 244.0 & 244.0 \\ 548.0 & 548.0 & 548.0 & 0.0 & 0.0 & 0.0 \\ 548.0 & 548.0 & 548.0 & 0.0 & 0.0 & 0.0 \\ 548.0 & 548.0 & 548.0 & 0.0 & 0.0 & 0.0 \end{bmatrix}$$

The activation levels of the MNs are then updated as follows:

$$A^{t+1}_{MN} = A^t_{MN} + W^T_{SN\rightarrow MN} * A^t_{SN} - \Theta_{MN}$$

where **A**MN is a vector of activation levels of MNs, and **θ**MN is a vector of thresholds of MNs.

The adjacency matrix representing the sensory feedback to the 13As is:

$$W_{SN \to 13A} = \begin{bmatrix} 526.0 & 0.0 \\ 526.0 & 0.0 \\ 526.0 & 0.0 \\ 153.0 & 48.0 \\ 153.0 & 48.0 \\ 153.0 & 48.0 \end{bmatrix}$$

And the **A**13A activation levels are updated:

$$A_{13A}^{t+1} = A_{13A}^{t} + W_{SN \to 13A}^{T} * A_{SN}^{t}$$

Activation levels are constrained as follows:

max(**A**13A)=2,500
max(**A**MN;**A**SN)=200

Two 13B nodes were added to inhibit the two 13 A nodes. The initial weights of the 13B -->13 A were –413 and –160.

## The excitatory network

The model network also needs an excitatory input. Because we do not know the upstream excitatory connections from the connectome, we created a 'black box' network that takes the simulated dust distribution as the input, and it outputs excitatory signal to the 13 A, 13B, and MNs. Other inputs include: the 'proprioceptive' inputs ('efference copy' from motor neurons), and the amount of dust accumulated on both legs. The 'black box' excitatory network consists of a recurrent neural network (RNN) as the hidden layer, with linear synapses and initially random weights.

The hidden RNN layer has 40 nodes. The input layer is the dust grid and the pixel values are the inputs. The output layer has 21 nodes connecting to the two 13B nodes, the two 12 A nodes, and the 6 MN nodes, *per one leg* (so 10 outputs per leg). The same excitatory network also feeds to the other leg's 13 A network, in the same manner. The remaining output node does not project anywhere (it is placed there for future model development where it could output the amplitude of exploratory noise injected into the 13 A network).

## The agent and the environment

Our model is composed of neuronal circuits embedded in a simple agent that acts on its environment. The agent has two 2D legs, corresponding to the two front legs of a fly, each of which is made of three points ('joints'): proximal, medial, and distal. The distal point (the one farthest from the 'body') can remove the 'dust' from the environment. The movement of the 'joints' (per leg) is controlled by three pairs of 'antagonistic muscles' affecting the three angles formed by the 'joints'. *Figure 6I* shows a frame of a movie where the legs are represented by the three points each.

The environment is composed of a Gaussian distribution of 'dust' around the agent (green pixels in *Figure 6I*). The means of the Gaussian are at the center of the 32 by 32 pixel grid (also the position of the 'root' of the front legs), so x=0; y=0. The variance = 5 pixels, in both directions. The maximal amount of dust (at the peak of the Gaussian) is 1.0 (0.99).

The agent can remove the 'dust' when the distal 'joint' sweeps over the environment with the minimum velocity of 1 pixel/frame. (So, if the 'joint' just stays at a given position, the dust is not getting removed. It has to move over it.)

As the grooming behavior is being performed, the 'dust' accumulates on the legs, reducing their ability to continue removing it from the grid (the 'body'). The 'leg cleaning' - removal of the accumulated dust occurs when the two legs are in proximity to each other (Euclidean distance <5 pixels). The 'leg cleaning' rate of dust removal from legs is the same as the body dust removal rate: 0.5/frame.

The three angles (per leg) are constrained. Distal and medial angles: 100° - 180°; proximal angle: 80° –120°.

## The fine-tuning of the synaptic weights

When we run the model with the default synaptic weights (see previous sections), nothing happens, that is the activation levels either saturate (reach the ceiling values shown above), fall to zero, or reach a steady value. The legs of the agent may move once or twice and then the model 'freezes'. One way of getting around this problem would be to add a 'CPG component' to the model, to drive the periodic excitatory inputs, thereby creating a baseline periodic activity and movements. We could then study the effects of the 13 A circuitry on these movements. However, we wanted to see if the model network could generate periodic movements all by itself.

The empirical weights used in the model are approximate, and we do not know the exact ratios between individual modeled weights (the assumption is that the number of biological synapses corresponds to the weights). So, we allow the modeled weights to vary (in value, but not in sign) and yet preserve the approximate ratios obtained from the empirical data. In other words, we are exploring a space of possible models that adhere to approximately the weight ratios obtained empirically. Specifically, the weights are allowed to vary +/- 20% and cannot change the sign – that is inhibitory neurons must remain inhibitory. (In future work, we may decrease this space of exploration to adhere even more closely to the empirical estimates. Conversely, we may increase the exploration space and observe all solutions to see how close our empirical estimates are to the global optimum.)

We thus constrain the model to the connectome as follows:

The connectome matrix reference **Wref** contains the empirical weights for a given neuron type (e.g. **W13A** ⟷ **13** A) and **Wmod** contains the weights from the evolving model (see below). First, we eliminate non-zero weights from the model (**Wmod**) that should be zero in the reference matrix **Wref**:

$$
M_{ij} = \begin{cases} 1 \ if \ W_{ref,ij} \neq 0 \\ 0 \ if \ W_{ref,ij} = 0 \end{cases}
$$

$$
W_{masked} = M \odot W_{raw}
$$

Because the absolute numbers in the empirical matrices (**Wref**) are meaningless for the modeling, we can rescale them—assuming that the ratios of the synaptic weights between the same types of neurons are preserved. So, each adjacency matrix connecting the same type of neurons is scaled separately. For example, **W13A → MN** is specifying the connections between 13As and MNs and is thus scaled separately from **W13A** ⟷ **13** A which specifies connections between 13As. Each **Wref** is scaled as shown:

$$
a_{raw} = max_{i,j} \left| W_{masked,ij} \right|, a_{ref} = max_{i,j} \left| W_{ref,ij} \cdot M_{ij} \right|
$$

$$
W_{ref}^{scaled} = \frac{a_{raw}}{a_{ref}} W_{ref}
$$

We then compute the tolerance bounds, where tolerance $\tau=0.2$:

$$
W_{min} = \left(1 - \tau\right) W_{ref}^{scaled}, W_{max} = \left(1+\tau\right) W_{ref}^{scaled}
$$

Finally, we enforce sign constraints and clip the model matrix **Wmod**:

$$
W_{signed,ij} = \left| W_{masked,ij} \right| \, sgn \left( W_{ref,ij} \right)
$$

$$
W_{mod,ij} = \begin{cases} min \left( max \left( W_{signed,ij}, W_{min,ij} \right), W_{max,ij} \right) & if \, M_{ij} = 1 \\ 0 & if \, M_{ij} = 0 \end{cases}
$$

To explore the space around our empirically estimated weights, we employed a genetic algorithm (GA) that we wrote (see the model code at https://github.com/PrimozRavbar/Inhibitory-circuits; *Ravbar, 2025*) Briefly, we evolved a population of 250–500 genomes containing the model parameters (weight matrices, thresholds) and hyper-parameters (max firing rates). The mutation rate was set to 0.1. The genomes were competing in cohorts of 6. At each epoch (each genome – agent – in the

cohort played 2 'games'), fitness rates were assigned and the strongest genomes were allowed to reproduce accordingly. *There was no cross-over.* Importantly, for every new generation, the mutated genomes were adjusted to fit the empirical weights within the tolerance limit of 0.2 (clipped, as described above). This way, the evolutionary exploration remained bounded throughout the process.

## Connectome analysis

### Neuronal reconstruction, lineage identification, and detection of neuronal partners

We used serial-section transmission electron microscopy (TEM) dataset of female adult *Drosophila* (FANC) *Phelps et al., 2021* to reconstruct 13 A and 13B hemilineages in the VNC. These neurons were identified in the EM volume based on their cell body clusters, arborization pattern, and nerve bundle entry positions into the ventral neuropil (*Lacin et al., 2019*; *Harris et al., 2015*; *Shepherd et al., 2016*) and comparison with light-level images and axonal tracts labeled with anti-Neuroglian.

13 A neurons cluster together and enter the VNC neuropil anteriorly through the ventrolateral position (*Lacin et al., 2019*; *Harris et al., 2015*; *Shepherd et al., 2016*; *Court et al., 2020*). 13B neurons have contralateral cell bodies and ipsilateral projections, with their axons entering the neuropil through the extreme ventral bundle (*Lacin et al., 2019*; *Harris et al., 2015*; *Shepherd et al., 2016*). Using confocal microscopy images of 13 A and 13B neurons marked with GFP and axonal tracts labeled with anti-Neuroglian for reference comparison, we located these neurons in the EM volume. We manually traced the main neural skeletons and later proofread automatic segmentations. Then neuronal IDs and cell body coordinates of each 13 A and 13B neurons are shown in *Supplementary file 1*.

Manual reconstructions of some of the 13 A and 13B neurons were initially performed in CATMAID (*Saalfeld et al., 2009*). Traced skeletons were then imported from CATMAID to Neuroglancer (*Maitin-Shepard et al., 2021*). We identified 13 A neurons and 13B neurons in the corresponding hemilineage bundles and proofread errors in the automated neuronal reconstructions (*Azevedo et al., 2024*). We fully proofread 62 13 A neurons (*Supplementary file 1*), 64 13B neurons in the right prothoracic segment (T1) of VNC, and 25/64 13B neurons in the left T1. We used the automated synapse detection to identify the downstream and upstream connections (*Azevedo et al., 2024*). We used various FANC packages (*Azevedo et al., 2024*) generously available to the community to generate upstream and downstream partner summary of all the 13 A and 13B neurons in R studio (available here).

### Connectivity matrix

To plot connectivity matrices between groups of neurons, we utilized Python libraries including pandas, networkx, and matplotlib. We created a directed graph using networkx to represent the connections, where presynaptic and postsynaptic neurons were added as nodes. The thickness and color of edges between nodes were determined by the strength of the connections and the type of presynaptic or postsynaptic neurons, respectively. Node colors were assigned based on the type of neurons, with specific colors denoting different subtypes of 13 A/B neurons and MNs. Finally, we generated the visualization using matplotlib. 13B to 13 A connections were manually added in *Figure 3B*. Leg schematic and MN to muscle connections were also manually added in Adobe Illustrator. (see the code at https://github.com/SyedDurafshan/Inhibitory-circuits-Source.git; *Syed, 2026*).

### Cosine similarity matrix

We computed the cosine similarity matrix of 13 A neurons based on their downstream motor connections in Python using the cosine_similarity from sklearn.metrics.pairwise for computing cosine similarities. A pivot table was created from the DataFrame, with neurons as rows (index) and their post-synaptic targets (post_id) as columns. The values in this table represented the weights of the connections. Duplicates were aggregated using the sum function, and missing values were filled with zeros. The cosine similarity between each pair of neurons was calculated using the cosine_similarity function. Cosine similarity is a measure that calculates the cosine of the angle between two vectors. In this context, each neuron is represented as a vector of its connectivity weights to downstream MNs. The cosine similarity value ranges from –1–1, where: 1 indicates that the vectors are identical. 0 indicates that the vectors are orthogonal (no similarity). –1 indicates that the vectors are diametrically opposed. This calculation resulted in a similarity matrix, where each entry (i, j) represents the cosine similarity between the connectivity profiles of neuron i and neuron j. This calculation resulted

in a similarity matrix, where each entry (i, j) represents the cosine similarity between the connectivity profiles of neuron i and neuron j. The resulting cosine similarity matrix was visualized using matplotlib. The matrix was displayed as a heatmap with a color gradient indicating the degree of similarity.

## Classification of 13A and 13B neurons based on morphology

We used NBLAST (*Azevedo et al., 2024*; *Costa et al., 2016*), a computational method to measure pairwise similarity between neurons based on their position and geometry to identify various subclasses within the hemilineages. We performed hierarchical clustering on pairwise NBLAST similarity scores computed using navis.nblast_allbyall(). The resulting similarity matrix was symmetrized by averaging it with its transpose and converted into a distance matrix using the transformation:

$$\text{distance} = (1 - \text{similarity}) \ \text{distance} = (1 - \text{similarity}) \ \text{distance} = (1 - \text{similarity})$$

This ensures that a perfect NBLAST match (similarity = 1) corresponds to a distance of 0.

Clustering was performed using Ward's linkage method (method='ward' in scipy.cluster.hierarchy. linkage), which minimizes the total within-cluster variance and is well-suited for identifying compact, morphologically coherent clusters. We did not predefine the number of clusters. Instead, clusters were visualized using a dendrogram, where branch coloring is based on the default behavior of scipy. cluster.hierarchy.dendrogram(). By default, this function applies a visual color threshold at 70% of the maximum linkage distance to highlight groups of similar elements. In our dataset, this corresponded to a linkage distance of approximately 1–1.5, which visually separated morphologically distinct neuron types (*Figure 2A*, *Figure 2—figure supplement 3A*). This threshold was used only as a visual aid and not as a hard cutoff for quantitative grouping. This classification was based on similarity scores and included left-right comparisons.

## Acknowledgements

We thank UCSB undergraduate members, especially Ethan Zhang, Yarah Meijer, Daniel Perry, Kaya Minami, and Allene Dang for experimental assistance and proofreading neurons. Sara Abraham, EZ, YM, and KM. contributed to manual behavioral labeling. Gabe Bello, Katelyn Ross, Kelly McDonald, Kai Thomas, Jada Moore, Yash Shah, Jinyi Dong, Mark Lu, Charliene Lien, Sofia Easton, William Jaber, Paige Gambetta, Maya Teitz, Dhruvi Dalwadi, Abdallah Samarah, Jonathan Carranza, Chandni Patel, Hana Nguyen, Inzar Khan, Garima Sehgal, Sydney Mauch, Yida Huang, Liz Kaslewicz, Nina Shenoy, Joseph Perliss, and Lindsay Easter proofread 13A/B neurons and their partners in the EM dataset. YM, KR, MT, KM, GB, YS, JM, LE led the EM team, with training provided by Li Guo, David McNeill, Ladann Kiassat, GB, EZ, and YM. We thank Wei-Chung Allen Lee, John Tuthill, and Jasper Phelps for access to the FANC connectome, the FANC community for generously sharing resources, and the Janelia and Cambridge groups for the MANC data. We are grateful to Akinao Nose and Shingo Yoshikawa for manuscript feedback. For fly strains, we thank James Truman, Haluk Lacin, Gerald Rubin, Benjamin White, Adam Claridge-Chang, and the Bloomington Drosophila Stock Center (NIH grant P40OD018537). This work was supported by NSF Career Award IOS-1943276 and NIH grant RF1NS132900.

## Additional information

### Funding

| Funder | Grant reference number | Author |
| --- | --- | --- |
| National Institutes of Health | NS132900 | Julie H Simpson |
| National Science Foundation | NSF1943276 | Julie H Simpson |

The funders had no role in study design, data collection and interpretation, or the decision to submit the work for publication.

## Author contributions
Durafshan Sakeena Syed, Conceptualization, Formal analysis, Investigation, Methodology, Writing – original draft, Writing – review and editing; Primoz Ravbar, Data curation, Software, Formal analysis, Methodology, Writing – review and editing; Julie H Simpson, Conceptualization, Supervision, Funding acquisition, Project administration, Writing – review and editing

## Author ORCIDs
Durafshan Sakeena Syed https://orcid.org/0000-0002-5729-0382
Primoz Ravbar https://orcid.org/0000-0001-7769-8687
Julie H Simpson https://orcid.org/0000-0002-6793-7100

Reviewer #1 (Public review): https://doi.org/10.7554/eLife.106446.4.sa1
Reviewer #3 (Public review): https://doi.org/10.7554/eLife.106446.4.sa2
Author response https://doi.org/10.7554/eLife.106446.4.sa3

## Additional files

### Supplementary files
Supplementary file 1. 13 A and 13B Neurons in the Front Leg Neuromere. The FANC identification numbers and coordinates, with the best matches in MANC and associated annotations, are listed for left and right neuromeres in the T1 segment.

MDAR checklist

### Data availability
Source data and Code for behavior and modeling is publicly available at: https://github.com/Primoz-Ravbar/Inhibitory-circuits (copy archived at *Ravbar, 2025*) and https://github.com/SyedDurafshan/Inhibitory-circuits-Source.git (copy archived at *Syed, 2026*).

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
