## [Editor Report · eLife Assessment]

Combining connectomics, optogenetics, behavioral analysis and modeling, this study delivers **important** findings on the role of inhibitory neurons in the generation of leg grooming movements in *Drosophila*. The results include **convincing** evidence that the identified neuronal populations are key in the generation of rhythmic leg movements, structured in distinct polysynaptic pathways articulating inhibition and disinhibition of antagonistic sets of motor neurons, as mapped from an electron microscopy volume of the ventral nerve cord, which orchestrate an alternation of flexion and extension. By analyzing limb kinematics upon experimentally silencing specific populations of premotor inhibitory neurons, together with computational modelling, the potential role of these neurons in rhythmic leg movement is shown. This work will be of interest to neuroscientists working in motor control and limbed locomotion.

---

## [Referee Report · Reviewer #1 (Public review)]

Summary:

Syed et al. investigate the circuit underpinnings for leg grooming in the fruit fly. They identify two populations of local interneurons in the right front leg neuromere of ventral nerve cord, i.e. 62 13A neurons and 64 13B neurons. Hierarchical clustering analysis identifies each 10 morphological classes for both populations. Connectome analysis reveals their circuit interactions: these GABAergic interneurons provide synaptic inhibition either between the two subpopulations, i.e. 13B onto 13A, or among each other, i.e. 13As onto other 13As, and/or onto leg motoneurons, i.e. 13As and 13Bs onto leg motoneurons. Interestingly, 13A interneurons fall into two categories with one providing inhibition onto a broad group of motoneurons, being called "generalists", while others project to few motoneurons only, being called "specialists". Optogenetic activation and silencing of both subsets strongly effects leg grooming. As well activating or silencing subpopulations, i.e. 3 to 6 elements of the 13A and 13B groups has marked effects on leg grooming, including frequency and joint positions and even interrupting leg grooming. The authors present a computational model with the four circuit motifs found, i.e. feed-forward inhibition, disinhibition, reciprocal inhibition and redundant inhibition. This model can reproduce relevant aspects of the grooming behavior.

Strengths:

The authors succeeded in providing evidence for neural circuits interacting by means of synaptic inhibition to play an important role in the generation of a fast rhythmic insect motor behavior, i.e. grooming of the body using legs. Two populations of local interneurons in the fruit fly VNC comprise four inhibitory circuit motifs of neural action and interaction: feed-forward inhibition, disinhibition, reciprocal inhibition and redundant inhibition. Connectome analysis identifies the similarities and differences between individual members of the two interneuron populations. Modulating the activity of small subsets of these interneuron populations markedly affects generation of grooming behavior thereby exemplifying their relevance. The authors carefully discuss strengths and limitations of their approaches and place their findings into the broader context of motor control.

Weaknesses:

Effects of modulating activity in the interneuron populations by means of optogenetics were conducted in the so-called "closed-loop" condition. This does not allow to differentiate between direct and secondary effects of the experimental modification in neural activity, as feedforward and feedback effects cannot be disentangled. To do so open loop experiments, e.g. in deafferented conditions, would be needed. Given that many members of the two populations of interneurons do not show one, but two or more circuit motifs, it remains to be disentangled which role the individual circuit motif plays in the generation of the motor behavior in intact animals.

Comments on revisions:

The authors have carefully revised the manuscript. I have no further suggestions or criticisms.

---

## [Referee Report · Reviewer #3 (Public review)]

Summary:

The authors set out to determine how GABAergic inhibitory premotor circuits contribute to the rhythmic alternation of leg flexion and extension during *Drosophila* grooming. To do this, they first mapped the ~120 13A and 13B hemilineage inhibitory neurons in the prothoracic segment of the VNC and clustered them by morphology and synaptic partners. They then tested the contribution of these cells to flexion and extension using optogenetic activation and inhibition and kinematic analyses of limb joints. Finally, they produced a computational model representing an abstract version of the circuit to determine how the connectivity identified in EM might relate to functional output. The study makes important contributions to the literature.

The authors have identified an interesting question and use a strong set of complementary tools to address it:

They analysed serial‐section TEM data to obtain reconstructions of every 13A and 13B neuron in the prothoracic segment. They manually proofread over 60 13A neurons and 64 13B neurons, then used automated synapse detection to build detailed connectivity maps and cluster neurons into functional motifs.

They used optogenetic tools with a range of genetic driver lines in freely behaving flies to test the contribution of subsets of 13A and 13B neurons.

They used a connectome-constrained computational model to determine how the mapped connectivity relates to the rhythmic output of the behavior.

Comments on revisions:

I appreciate that the authors have updated the GitHub repository to include the model and analysis code. Still lacking is: for the authors to explicitly separate empirical findings from modelling inferences in the text, and a supplemental table to make it clear which cell types are included. I should also point out that the code lacks annotations necessary for the results to be reproduced and the model to be reused.

---

## [Author Response]

The following is the authors’ response to the previous reviews.

**Public Reviews:**

**Reviewer #1 (Public review):**
Summary:Syed et al. investigate the circuit underpinnings for leg grooming in the fruit fly. They identify two populations of local interneurons in the right front leg neuromere of ventral nerve cord, i.e. 62 13A neurons and 64 13B neurons. Hierarchical clustering analysis identifies each 10 morphological classes for both populations. Connectome analysis reveals their circuit interactions: these GABAergic interneurons provide synaptic inhibition either between the two subpopulations, i.e. 13B onto 13A, or among each other, i.e. 13As onto other 13As, and/or onto leg motoneurons, i.e. 13As and 13Bs onto leg motoneurons. Interestingly, 13A interneurons fall into two categories with one providing inhibition onto a broad group of motoneurons, being called "generalists", while others project to few motoneurons only, being called "specialists". Optogenetic activation and silencing of both subsets strongly effects leg grooming. As well activating or silencing subpopulations, i.e. 3 to 6 elements of the 13A and 13B groups has marked effects on leg grooming, including frequency and joint positions and even interrupting leg grooming. The authors present a computational model with the four circuit motifs found, i.e. feed-forward inhibition, disinhibition, reciprocal inhibition and redundant inhibition. This model can reproduce relevant aspects of the grooming behavior.Strengths:The authors succeeded in providing evidence for neural circuits interacting by means of synaptic inhibition to play an important role in the generation of a fast rhythmic insect motor behavior, i.e. grooming. Two populations of local interneurons in the fruit fly VNC comprise four inhibitory circuit motifs of neural action and interaction: feed-forward inhibition, disinhibition, reciprocal inhibition and redundant inhibition. Connectome analysis identifies the similarities and differences between individual members of the two interneuron populations. Modulating the activity of small subsets of these interneuron populations markedly affects generation of the motor behavior thereby exemplifying their important role for generating grooming. The authors carefully discuss strengths and limitations of their approaches and place their findings into the broader context of motor control.

We thank the reviewer for their thoughtful and constructive evaluation of our work.

Weaknesses:Effects of modulating activity in the interneuron populations by means of optogenetics were conducted in the so-called closed-loop condition. This does not allow to differentiate between direct and secondary effects of the experimental modification in neural activity, as feedforward and feedback effects cannot be disentangled. To do so open loop experiments, e.g. in deafferented conditions, would be important. Given that many members of the two populations of interneurons do not show one, but two or more circuit motifs, it remains to be disentangled which role the individual circuit motif plays in the generation of the motor behavior in intact animals.

Our optogenetic experiments show a role for 13A/B neurons in grooming leg movements – in an intact sensorimotor system - but we cannot yet differentiate between central and reafferent contributions. Activation of 13As or 13Bs disinhibits motor neurons and that is sufficient to induce walking/grooming. Therefore, we can show a role for the disinhibition motif.

Proprioceptive feedback from leg movements could certainly affect the function of these reciprocal inhibition circuits. Given the synapses we observe between leg proprioceptors and 13A neurons, we think this is likely.

Our previous work (Ravbar et al 2021) showed that grooming rhythms in dusted flies persist when sensory feedback is reduced, indicating that central control is possible. In those experiments, we used dust to stimulate grooming and optogenetic manipulation to broadly silence sensory feedback. We cannot do the same here because we do not yet have reagents to separately activate sparse subsets of inhibitory neurons while silencing specific proprioceptive neurons. More importantly, globally silencing proprioceptors would produce pleiotropic effects and severely impair baseline coordination, making it difficult to distinguish whether observed changes reflect disrupted rhythm generation or secondary consequences of impaired sensory input. Therefore, the reviewer is correct – we do not know whether the effects we observe are feedforward (central), feedback sensory, or both. We have included this in the revised results and discussion section to describe these possibilities and the limits of our current findings.

Additionally, we have used a computational model to test the role of each motif separately and we show that in the results.

Comments on revisions:The careful revision of the manuscript improved the clarity of presentation substantially.
**Reviewer #2 (Public review):**
Summary:This manuscript by Syed et al. presents a detailed investigation of inhibitory interneurons, specifically from the 13A and 13B hemilineages, which contribute to the generation of rhythmic leg movements underlying grooming behavior in *Drosophila*. After performing a detailed connectomic analysis, which offers novel insights into the organization of premotor inhibitory circuits, the authors build on this anatomical framework by performing optogenetic perturbation experiments to functionally test predictions derived from the connectome. Finally, they integrate these findings into a computational model that links anatomical connectivity with behavior, offering a systems-level view of how inhibitory circuits may contribute to grooming pattern generation.Strengths:(1) Performing an extensive and detailed connectomic analysis, which offers novel insights into the organization of premotor inhibitory circuits.(2) Making sense of the largely uncharacterized 13A/13B nerve cord circuitry by combining connectomics and optogenetics is very impressive and will lay the foundation for future experiments in this field.(3) Testing the predictions from experiments using a simplified and elegant model.

Thank you for the positive assessment of our work.

Weaknesses:(1) In Figure 4-figure supplement 1, the inclusion of walking assays in dusted flies is problematic, as these flies are already strongly biased toward grooming behavior and rarely walk. To assess how 13A neuron activation influences walking, such experiments should be conducted in undusted flies under baseline locomotor conditions.

We agree that there are better ways to assay potential contributions of 13A/13B neurons to walking. We intended to focus on how normal activity in these inhibitory neurons affects coordination during grooming, and we included walking because we observed it in our optogenetic experiments and because it also involves rhythmic leg movements. The walking data is reported in a supplementary figure because we think this merits further study with assays designed to quantify walking specifically. We will make these goals clearer in the revised manuscript and we are happy to share our reagents with other research groups more equipped to analyze walking differences.

(2) Regarding Fig 5: The 70ms on/off stimulation with a slow opsin seems problematic. CsChrimson off kinetics are slow and unlikely to cause actual activity changes in the desired neurons with the temporal precision the authors are suggesting they get. Regardless, it is amazing the authors get the behavior! It would still be important for authors to mention the optogentics caveat, and potentially supplement the data with stimulation at different frequencies, or using faster opsins like ChrimsonR.

We were also intrigued by the behavioral consequences of activating these inhibitory neurons with CsChrimson. We appreciate the reviewer’s point that CsChrimson’s slow off-kinetics limit precise temporal control. To address this, we repeated our frequency analysis using a range of pulse durations (10/10, 50/50, 70/70, 110/110, and 120/120 ms on/off) and compared the mean frequency of proximal joint extension/flexion cycles across conditions. We found no significant difference in frequency (LLMS, p > 0.05), suggesting that the observed grooming rhythm is not dictated by pulse period but instead reflects an intrinsic property of the premotor circuit once activated. We now include these results in ‘Figure 5—figure supplement 1’ and clarify in the text that we interpret pulsed activation as triggering, rather than precisely pacing, the endogenous grooming rhythm. We continue to note in the manuscript that CsChrimson’s slow off-kinetics may limit temporal precision. We will try ChrimsonR in future experiments.

Overall, I think the strengths outweigh the weaknesses, and I consider this a timely and comprehensive addition to the field.
**Reviewer #3 (Public review):**
Summary:The authors set out to determine how GABAergic inhibitory premotor circuits contribute to the rhythmic alternation of leg flexion and extension during *Drosophila* grooming. To do this, they first mapped the ~120 13A and 13B hemilineage inhibitory neurons in the prothoracic segment of the VNC and clustered them by morphology and synaptic partners. They then tested the contribution of these cells to flexion and extension using optogenetic activation and inhibition and kinematic analyses of limb joints. Finally, they produced a computational model representing an abstract version of the circuit to determine how the connectivity identified in EM might relate to functional output. The study makes important contributions to the literature.The authors have identified an interesting question and use a strong set of complementary tools to address it:They analysed serial‐section TEM data to obtain reconstructions of every 13A and 13B neuron in the prothoracic segment. They manually proofread over 60 13A neurons and 64 13B neurons, then used automated synapse detection to build detailed connectivity maps and cluster neurons into functional motifs.They used optogenetic tools with a range of genetic driver lines in freely behaving flies to test the contribution of subsets of 13A and 13B neurons.They used a connectome-constrained computational model to determine how the mapped connectivity relates to the rhythmic output of the behavior.
**Recommendations for the authors:**

**Reviewer #1 (Recommendations for the authors):**

I still have the following specific suggestions and questions, which need the attention of the authors:

P5, 2nd para, li 1: shouldn't "(Figures 1E and 1E')" be (Figures 1G and 1H)?P7, last para, li 3: shouldn't "(Figures 2C and 2D)" be (Figures 2A and 2B)?P19, para 2, last 2li: "...we observe that optogenetic activation......triggers grooming movements." I could not find the place in the text or a figure, where this was reported or shown. Please specifyP19, last para: "... shows that 13A neurons can generate rhyhtmic movements....." Given that the experiments were conducted in closed-loop, i.e. including the loop through the leg and its movements, the following formulation appears more justified: "....shows that 13A neurons significantly contribute to the generation of rhythmic movements,....."P28, para 1, li 3 from bottom: "...themselves, rather than solely between antagonistsic motor neurons." While the authors are correct that in the stick insect and locust alternating inhibitory synaptic drive to flexor and extensor motoneurons has been shown to underly alternating activity of these two antagonistic motoneuron pools the previous studies have not shown or claimed that these synaptic inputs arise from direct interactions between these motoneuron pools. Based on this this text should be moved to the part "feed-forward inhibition" on page 27.P28: "redundant inhibition": this motif has been shown to be instrumental in the locust flight CPG, e.g. Robertson & Pearson, 1985, Fig. 16.P28: "reciprocal inhibition" The reviewer agrees with the authors that this motif has been shown for the mouse spinal cord, but also for other CPGs in vertebrates and invertebrates, e.g. clione, leech, xenopus - see the initial comment "(3) Intro and Discussion"

Thank you, we have incorporated the suggested corrections and clarifications into the revised manuscript.

**Reviewer #2 (Recommendations for the authors):**
I'm satisfied with the revised version
**Reviewer #3 (Recommendations for the authors):**
The authors have made a substantial effort to address my original points. They corrected the title, expanded Discussion and Methods sections, reran statistical tests using mixed models, added modelling clarifications and constraints, and fixed or removed confusing figure panels. Those changes have improved clarity and reduced some of the claims that I thought were exaggerated.That said, some of my concerns remain only partially addressed, which could be fixed with relatively small tweaks. The authors should:(1) Explicitly separate empirical findings from modelling inferences throughout the manuscript, including the Abstract, Results and Discussion (i.e., label claims of "intrinsic rhythmogenesis" as model-based inferences, not direct experimental demonstrations)(2) Provide supplemental information on modelling to quantify the role of the black-box input (e.g., quantitative coordination/phase/frequency metrics for full model vs constant-input vs no black box), show pre- vs post-fine-tuning weight changes and the exact tuning constraints/optimization details (I could not find these details)(3) To ensure results are reproducible, provide a supplemental table mapping each split line to EM-identified neuron(s) with NBLAST/morphological scores for each match;(4) Fully document the statistical models (exact LMM/GLMM formulas, software/packages, etc);(5) Deposit model code, trained weights and analysis scripts in a public repository.

We have updated the GitHub repository with the full statistical analysis documentation and model code, including trained weights and scripts.